# Fluid mechanics of luminal transport in actively contracting endoplasmic reticulum

**Pyae Hein Htet[1], Edward Avezov[2]\*, Eric Lauga[1]\***

[1]Department of Applied Mathematics and Theoretical Physics, University of Cambridge, Cambridge, United Kingdom; [2]UK Dementia Research Institute at University of Cambridge, Department of Clinical Neurosciences, University of Cambridge, Cambridge, United Kingdom

### eLife Assessment

This work explores the physical principles underlying fluid flow and luminal transport within the endoplasmic reticulum. Its **important** contribution is to highlight the strong physical constraints imposed by viscous dissipation in nanoscopic tubular networks. In particular, the work presents **convincing** evidence based on theoretical analysis that commonly discussed mechanisms such as tubular contraction are unlikely to be at the origin of the observed transport velocities. As such, it will be of relevance to cell biologists and physicists interested in organelle dynamics. As this study is solely theoretical and deals with order of magnitude estimates, its main conclusions await experimental validation.

**\*For correspondence:**
ea347@medschl.cam.ac.uk (EA);
e.lauga@damtp.cam.ac.uk (EL)

**Competing interest:** The authors declare that no competing interests exist.

**Abstract** The endoplasmic reticulum (ER), the largest cellular compartment, harbours the machinery for the biogenesis of secretory proteins and lipids, calcium storage/mobilisation, and detoxification. It is shaped as layered membranous sheets interconnected with a network of tubules extending throughout the cell. Understanding the influence of the ER morphology dynamics on molecular transport may offer clues to rationalising neuro-pathologies caused by ER morphogen mutations. It remains unclear, however, how the ER facilitates its intra-luminal mobility and homogenises its content. It has been recently proposed that intra-luminal transport may be enabled by active contractions of ER tubules. To surmount the barriers to empirical studies of the minuscule spatial and temporal scales relevant to ER nanofluidics, here we exploit the principles of viscous fluid dynamics to generate a theoretical physical model emulating in silico the content motion in actively contracting nanoscopic tubular networks. The computational model reveals the luminal particle speeds, and their impact in facilitating active transport, of the active contractile behaviour of the different ER components along various time–space parameters. The results of the model indicate that reproducing transport with velocities similar to those reported experimentally in single-particle tracking would require unrealistically high values of tubule contraction site length and rate. Considering further nanofluidic scenarios, we show that width contractions of the ER's flat domains (perinuclear sheets) generate local flows with only a short-range effect on luminal transport. Only contractions of peripheral sheets can reproduce experimental measurements, provided they are able to contract fast enough.

## Introduction

The mammalian endoplasmic reticulum (ER) is the single largest intracellular structure (see sketch in *Figure 1a*). The organelle is made up of membranous sheets interconnected with the nuclear envelope and branching out into a planar network of tubules extending throughout the cell periphery

**Figure 1.** Sketch of the cellular geometry with nomenclature of the subcellular structures discussed in the paper. (**a**) Cross-section of cell showing nucleus and endoplasmic reticulum (ER) (adapted from image in public domain). (**b**) Cut through cross-section of the tubular ER network at the edge of cell. (**c**) Sketch of the contraction and expansion of the tubular junctions (3D view and cross-section); contractions leads to flow leaving the junction into the network while expansions lead to flow leaving the network and entering the junction. (**d**) Contraction and expansion of the peripheral sheets. (**e**) Contraction and expansion of the tubules driven by pinching (3D view and cross-section). (**f**) Contraction and expansion of the perinuclear sheets.

Panel A adapted from image in public domain (Ruiz Villarreal, 2006).

(*Figure 1b*, *Voeltz et al., 2002*). The ER dynamics on a second scale include the cytoskeleton-assisted tubular network restructuring (*Lee and Chen, 1988*; *Chambers et al., 2022*) and an interconversion between two distinct forms, including a narrower form covered by membrane curvature-promoting proteins (*Wang et al., 2022*). The ER morphology and its dynamics presumably enable and facilitate its functions: the ER is responsible for the production, maturation, and quality-controlled folding of secretory and membrane proteins, which constitute approximately a third of the cell's proteome (*Ghaemmaghami et al., 2003*). The organelle's membranes also harbour the lipid biosynthesis

machinery, while its lumen stores calcium. The contiguous nature of the ER is believed to ensure an efficient delivery of all these components across the cell periphery. In particular, ER luminal continuity and transport were demonstrated to kinetically limit calcium delivery for local release (*Crapart et al., 2024*). The sensitivity of neurons with long axonal extensions to ER defects in ER morphogens suggests that a perturbed ER transport may link ER integrity and neurodegeneration. Such a link might help explain why mutations in genes involved in ER shaping cause neuronal diseases, including motor neuron degeneration of hereditary spastic paraplegia (*Blackstone et al., 2011*), sensory neuropathy (*Kornak et al., 2014*) and retinitis pigmentosa (*Arno et al., 2016*).

Timely transport of the content within the ER is therefore integral to the function of the cell. The geometry and dimensions of several cell types with extensive ER-containing projections (e.g. neurons and astrocytes) pose a kinetic challenge for material distribution with physiological timing. These considerations predict the need for an active luminal transport to ensure timely material homogenisation across the vast ER. Empirically, the active nature of the ER luminal transport is suggested by a series of observations which include the sensitivity of Green fluorescent protein (GFP) bulk mobility (measured by Fluorescence recovery after photobleaching (FRAP) and photoactivation) to ATP depletion (*Nehls et al., 2000*; *Holcman et al., 2018*). However, these bulk fluorescence intensity dynamics techniques do not provide information on transport mode. By default, the intensity dynamics were historically fitted to diffusion models and the mobility kinetics was often expressed in terms of effective diffusion coefficients. Measurements of single-particle motion and chasing locally photoactivated luminal protein marker over distances, circumvent this limit and indicate inconsistencies with molecular diffusion (*Konno et al., 2024*).

However, the mechanism for generating ER luminal flows remains unclear. Understanding the mode of material exchange across the organelle is crucial for rationalising the ER shaping defect-related neuronal pathologies (*Blackstone, 2018*), identifying factors controlling ER transport and informing the development of ER transport modulation approaches with health benefits. Based on ER marker velocity fluctuations measured in single-particle tracking and the detection of transient narrowing points in the tubules by improved super-resolution and electron microscopy (*Holcman et al., 2018*), it has been postulated that these active flows may result from the stochastic contractility (pinching and unpinching) of ER tubules at specific locations along their lengths (see sketch in *Figure 1e*); other plausible mechanisms for flow generation were also considered. However, measurements for testing this pinching hypothesis are currently inaccessible, due to limitations in space–time resolution of live cell microscopy; the live cell-compatible super-resolution techniques achieve resolution of ~80 nm at the relevant speed, while the tubular radius is estimated in the range 30–60 nm (*Gao et al., 2019*; *Schroeder et al., 2019*; *Konno et al., 2024*). Improvement in resolution currently can only be achieved by trading off speed. To circumvent these experimental difficulties, in the current study we use mathematical modelling to quantitatively analyse the relevant scenarios of actively contractility-driven flows and to explore how various sets of spatiotemporal parameters of ER contractility may produce flows facilitating solute transport in quantitative agreement with experimental measurements.

We illustrate as a simple schematic in *Figure 1* the different contractility mechanisms in the different regions of the ER and where they are located in relation to the cell centre/nucleus. The ER close to the cell nucleus is geometrically complex, consisting of stacks of perinuclear sheets (*Figure 1a*). Away from the nucleus, the ER geometry simplifies considerably and the ER at the cell periphery is comprised of a planar network of tubules (*Figure 1b*). In the current work, we study flows and transport in this planar, tubular region; this is also the region of the ER network in which single-particle tracking measurements were carried out in *Holcman et al., 2018*. We consider as potential driving mechanisms for the observed solute transport the contractility of components located in the same planar tubular region, namely, tubular junctions (*Figure 1c*), peripheral sheets (*Figure 1d*), and tubules (*Figure 1e*), as well the contractility of perinuclear sheets (*Figure 1f*) located closer to the nucleus.

First, assuming that the flow is driven by tubule contractions (shown schematically in *Figure 1e*), we construct a physical and mathematical model of the ER network and solve it for the flows inside the network (description in Materials and methods). We use our model to carry out numerical simulations to study the motion of Brownian particles carried by these flows and show that the tubule pinching hypothesis is not supported by the results of our model (Tubule contractility-driven ER luminal motion yields inadequate transport kinetics), a result independent of the network geometry (Tubule pinching-induced transport is network-geometry independent and fails to facilitate luminal homogenisation).

The failure of active pinching to drive strong flows can be rationalised theoretically by deriving a rigorous upper-bound on the rate of transport induced by a single pinch (Theoretical analysis of advection due to a single pinch explains weak pinching-induced transport), including possible coordination mechanisms (Marginal addition of coordinated contractility to luminal transport). Only by increasing both the length of pinches and their rate beyond admissible values can we produce particle speeds in agreement with measurements (A combination of high frequency and pinch length is required to replicate experimental particle speeds). We then explore in (Luminal transport kinetics derived from contractile ER tubular junctions and sheets) two different hypotheses as possible explanations for the active ER flows, first the pinching of the junctions between tubules (*Figure 1c*) and then the pinching of the two types of ER sheets, perinuclear (*Figure 1f*) and peripheral (*Figure 1d*). We investigate the conditions under which these contractility mechanisms would be consistent with the experimental measurements of active transport in *Holcman et al., 2018*.

The question of how the ER homogenises its content across cell expanses is an open problem which extends to the fundamentals of cell biology. Even though it is highly debated in the field, it is extremely difficult to study due to the enormous technical limits of directly observing intraorganellar nanofluidics. Thus, our study represents a meaningful effort to break through this impasse by conducting a meticulous analysis of nanofluidic scenarios. The findings we present constitute the best currently possible endeavour to shed light on this challenging problem. Our results do suggest that the biological origin of solute transport in ER networks remains open and call for extensive empirical exploration of the alternative mechanisms for flow generation.

## Results

### Tubule contractility-driven ER luminal motion yields inadequate transport kinetics

To assess the kinetics of particle motion in the lumen of tubular structures, detailed in *Figure 1*, in response to their contractility, we generate an in silico simulation model of the process. The model incorporates local calculations for the low Reynolds number hydrodynamics of a contracting tubule, assuming in the first instance the no-slip boundary conditions at the tubule walls (i.e. Poiseuille hydrodynamics), into a global analysis of the flows throughout the network geometry, by using Kirchhoff's laws and standard graph theoretical results (see Materials and methods for details).

We initially implement the model's numerical simulations of particle transport in a reconstructed ER network of a COS-7 cell (*Holcman et al., 2018*) (which we label C0, see Network modelling) with tubules locally contracting stochastically according to the spatiotemporal parameters suggested by microscopy measurements (*Holcman et al., 2018*). We will therein refer to these contractions, illustrated in *Figure 1e*, as 'pinching', with the relevant parameters being the duration and frequency of pinch events, and the length that the pinch sites occupy along the tubule (for details of the pinching kinematics, see Materials and methods, Pinch modelling). An estimate reveals that these pinches are indeed afforded by biologically realistic forces, of the order of 30 pN (see Materials and methods, Estimate of forces required for pinches).

The fluid flows in the edges of the network (model in Hydrodynamic modelling computed as detailed in Solving the hydrodynamic network model) reveal a rapid direction alternation of luminal currents (on average with a frequency of approximately 50Hz), as reflected in the changes of the axial velocity sign (*Figure 2a*), with an average flow speed of 1.3μm/s (see also *Video 1*). Furthermore, the resulting instantaneous speed distribution of Brownian particles advected by these flows (methodology in Simulating particle transport) is considerably shifted relative to the experimental counterpart (*Holcman et al., 2018*) towards lower values (*Figure 2b*). Similarly, the distribution of average edge traversal speeds (defined precisely in Data processing: instantaneous speeds and average edge traversal speeds) from our simulations (mean 4.4μm/s, solid blue line in *Figure 2c*) is lower than the experimentally measured speed distribution (mean 45μm/s; *Figure 2c*, inset) by an order of magnitude.

Moreover, the results for all measures of transport under pure diffusion, in the absence of pinching-induced flows, are virtually indistinguishable from those where the pinching-driven flows are included (see *Figure 2b, c*). Within the framework of transport theory, this conclusion can be rationalised by estimating the relevant Péclet numbers that measure the relative importance of advection and

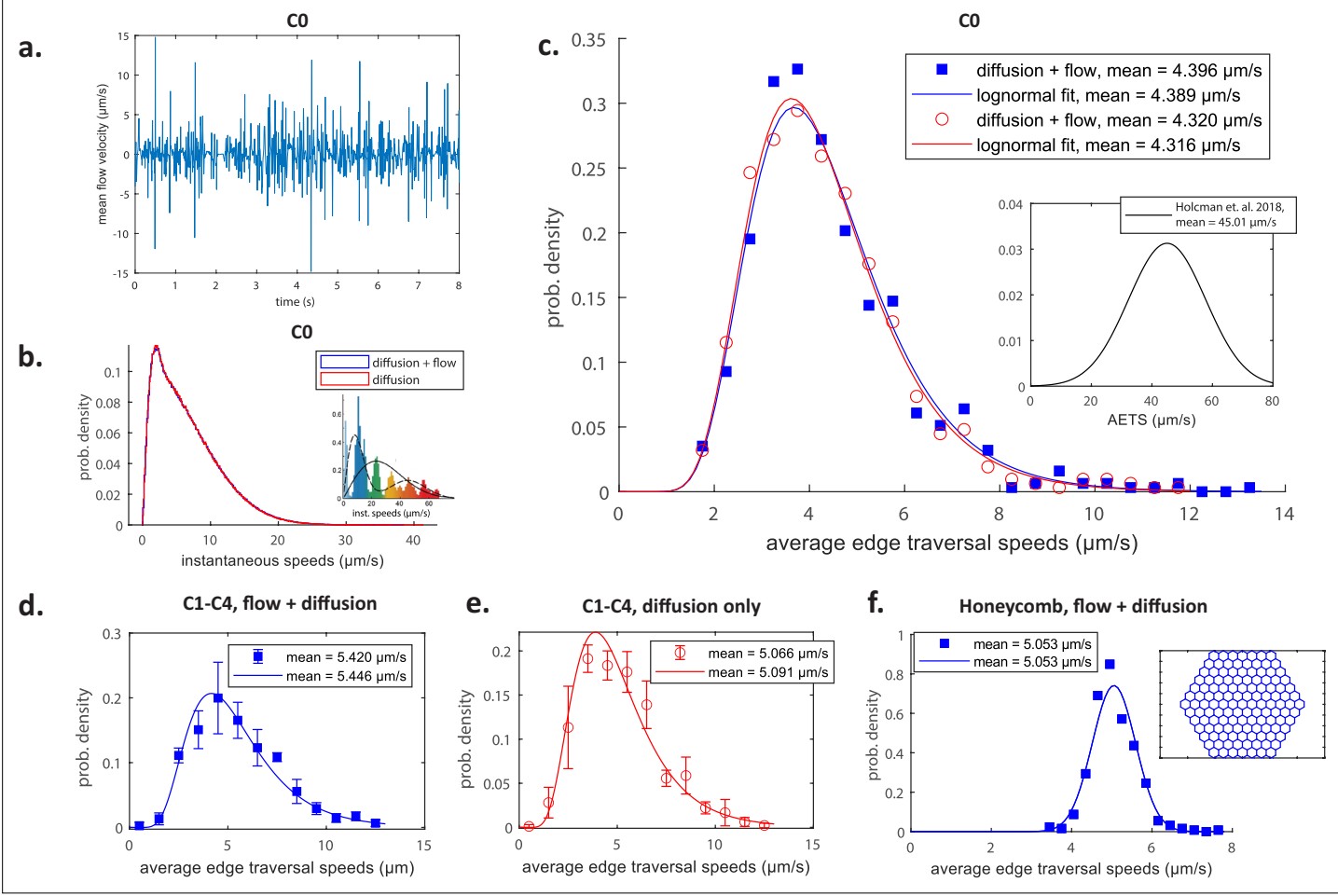

**Figure 2.** A quantitative test of the pinching-tubule hypothesis. (**a**) Cross-sectionally averaged flow velocities in a typical edge as obtained in our simulations. Histograms of instantaneous speeds (**b**) and edge traversal speeds (**c**) using data from simulations in the C0 network with flow (blue) and with just diffusion (red). The insets in (**b**) and (**c**) illustrate the distributions of instantaneous speeds and average edge traversal speeds, respectively, as experimentally measured in *Holcman et al., 2018*. The symbols indicate the values taken by the probability mass function and the curves are lognormal distributions fitted to all average edge traversal speeds obtained. Histograms of average edge traversal speeds obtained from simulations in networks C1–C4 from *Figure 9d* with flow (**d**) and only diffusion (**e**) and from simulations in the regular honeycomb network with active flows (**f**). The inset in (**f**) illustrates the honeycomb geometry. Points indicate mean ± 1 SD over the four networks (C1–C4) of normalised frequencies in each speed range; curves are log-normal (**d–f**) or normal (**f**) distributions fitted to all average edge traversal speeds for each set of pinch parameters. The means of the original simulation results and of the fitted distributions are indicated in the legends in each of (**c–f**).

diffusion. Using the average value $\bar{U} \sim 1.3$ μm/s of the mean flow speeds over time and edges as a velocity scale, we may estimate a mean Péclet number as $\mathrm{Pe} \sim \bar{U}R/D$. Using $R = 30$ nm and the measured diffusivity $D \approx 0.6$ μm² s⁻¹ (*Holcman et al., 2018*), this leads to the estimate $\mathrm{Pe} \sim 0.07$. Flows affect transport for Péclet numbers above order-one values (*Leal, 2007*), and therefore the pinching-driven flows have a negligible influence on the transport inside the ER network. In order for fluid motion to have a noticeable effect, one would thus need flows to be either stronger, or sustained in one direction for longer. The chaotic flows produced by the pinching events with stochastic parameters suggested by resolution-limited microscopy (*Holcman et al., 2018*) appear too weak to generate fast, non-diffusive edge traversals.

Importantly, our conclusions remain unchanged upon relaxing the no-slip boundary conditions, an important point to consider since the tubule membranes themselves could be moving in response to the nanoscale fluid flow. This can be modelled by introducing a finite slip boundary condition to couple the membrane-bound lipids with luminal flows (see Materials and methods, Incorporating slip boundary conditions for details). Our simulations show that slip boundary conditions have virtually no effect on the average edge traversal speeds (*Figure 3a*). Physically, while a non-zero wall slip

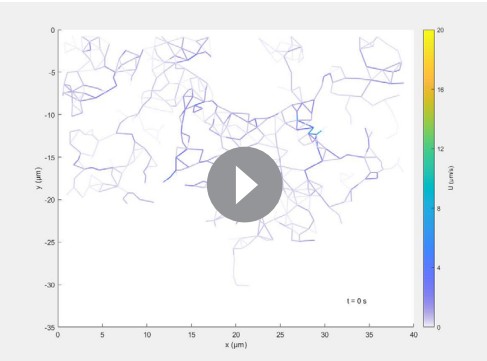

**Video 1.** Flows in an active C0 network pinching with the original pinch parameters. Edges are colour-coded with magnitude of instantaneous flow.
https://elifesciences.org/articles/93518/figures#video1

does change the shape of the flow profiles in the tubules (*Figure 3b*), it does not modify the cross-sectionally averaged flow speeds which directly affect global particle transport. These results justify our use of Poiseuille hydrodynamics (no-slip boundary conditions) throughout this work.

## Tubule pinching-induced transport is network geometry independent and fails to facilitate luminal homogenisation

To estimate the contribution of the network geometry to the outcomes of our transport simulations, we compare results across four different ER structures, which we label C1–C4. We illustrate these reconstructed networks along with the source data in Figure 9. The distributions of the average edge traversal speeds appear insensitive to ER structure variations for both pinching-induced and exclusively diffusional transport. This is reflected in the small deviations from the mean of the data points averaged across the different structures (*Figure 2d, e*). Furthermore, pinching-induced flows inside a regular honeycomb network (*Figure 2f*, inset) with a typical ER edge length (1 μm) appear to be within a reasonable variance compared to the natural networks. Therefore, this mathematical idealisation of the ER network geometry can be used for exploring the consequences of the network ultrastructure contractility on transport kinetics in a standardised manner.

To test the effectiveness of the particle velocities for facilitating luminal material exchange across the ER, we track homogenisation kinetics by measuring intermixing of particles of two distinct colours equally seeded in each half of a honeycomb network at $t = 0$ (*Figure 4*; see also *Video 2* for the flows inside such a network driven by pinches with the parameters from *Holcman et al., 2018*). The measure of homogenisation is given by the variance $\mathrm{Var}(\phi(t)) \equiv \mathrm{Var}(n_b(t) - n_r(t))$ over 20 regions of the network (*Figure 4a*, horizontal lines) of the difference between the numbers $n_b$ (blue) and $n_r$ (red) of particles of each colour in region; note $\mathrm{Var} = 0$ represents perfect homogeneity. The measures of mixing over time for pure diffusion and active transport with parameters (*Holcman et al., 2018*) from *Figure 2* show a nearly complete overlap (*Figure 4e*; see also *Video 3* and *Video 4*), distinct from faster mixing under stronger flows in a network driven by pinches whose lengths are increased to their

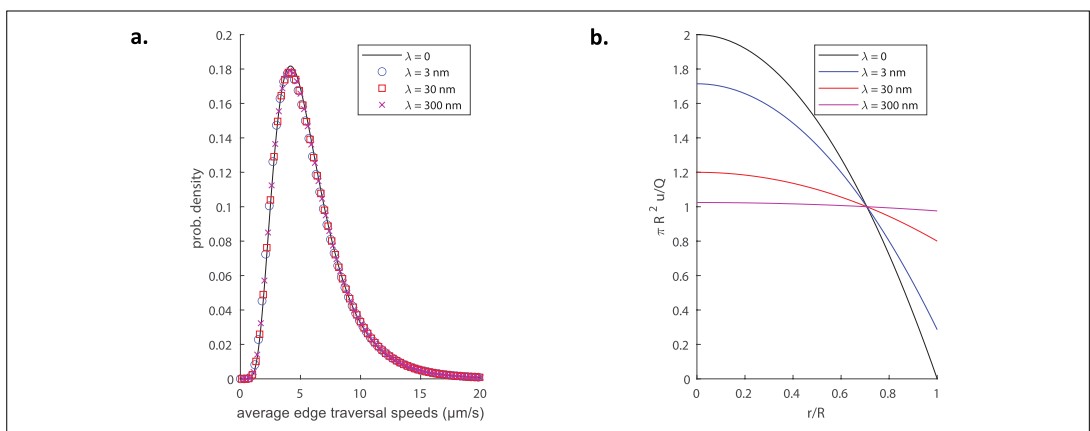

**Figure 3.** Impact of non-zero slip length on transport and flow. (**a**) Distributions of average edge traversal speeds in simulations of a C1 network pinching with the original pinch parameters as measured in *Holcman et al., 2018* and used in *Figure 2*, for different slip lengths $\lambda = 0, 3, 30,$ and 300 nm. (**b**) Longitudinal flow profile $u(r)$ inside a cylindrical tubule for different slip lengths, all with the same volume flux $Q$; an increase of the slip length leads to a redistribution of the flow in the cross-section.

**a.** t = 0 s

**b.** no flow, t = 3 s

**c.** original, t = 3 s

**d.** pinches $L_{max}$, 10x rate, t = 3 s

**e.**

Legend:
— no flow
— original
— pinches $L_{max}$, 10x rate

**Figure 4.** Mixing by active pinching flows. (**a**) Initial configuration of blue and red particles in honeycomb network. The strips used to quantify mixing are illustrated in black dotted line. The configuration after $t = 3$ s of mixing in a passive network with no flow (**b**), an active network pinching with the original pinch parameters (**c**), and an active network pinching with maximally long pinches at 10 times the original rates (**d**). (**e**) The measure of mixing $\mathrm{Var}(\phi(t))$ against $t$ for the passive network (blue), the network pinching with the original parameters (red), and the network pinching with maximally long and 10× faster pinches (yellow).

maximum possible value (i.e. the length of the tubule) and which are in addition sped up by a factor of 10 (see *Video 5*; see also (A combination of high frequency and pinch length is required to replicate experimental particle speeds) for a discussion of

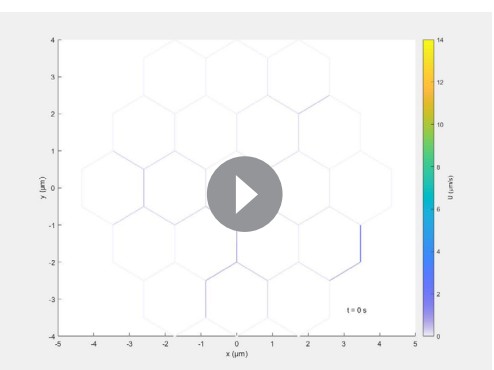

**Video 2.** Flows in an active honeycomb network pinching with the original pinch parameters. Edges are colour-coded with magnitude of instantaneous flow.
https://elifesciences.org/articles/93518/figures#video2

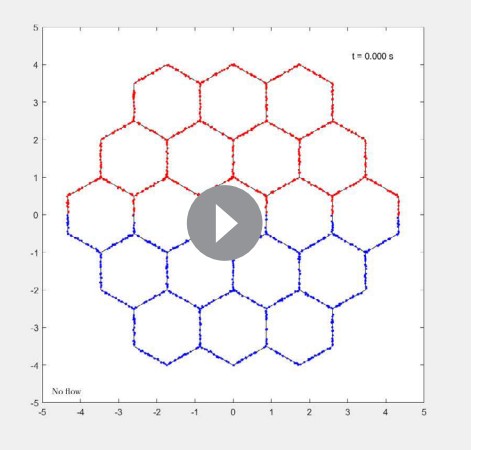

**Video 3.** Mixing in time in a passive honeycomb network with no flow.
https://elifesciences.org/articles/93518/figures#video3

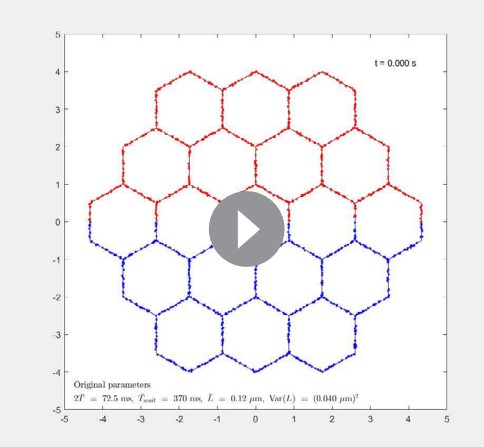

**Video 4.** Mixing in time in an active honeycomb network pinching with the original pinch parameters.
https://elifesciences.org/articles/93518/figures#video4

the effects of artificially strong pinch parameters on average edge traversal speeds). This indicates that the presumed biological pinching parameters would be inadequate to facilitate ER luminal material exchange.

## Theoretical analysis of advection due to a single pinch explains weak pinching-induced transport

The slow luminal transport driven by the pinching-induced flows is intrinsically linked to the volume of fluid expelled by each pinch during a contraction. The fundamental reason underlying the weak pinching-induced transport is that individual pinches are very weak generators of flow; even in the best possible configuration, the volumes of fluid pushed by each pinch are too small to impact luminal transport. Specifically, in Materials and methods (Advection due to a single pinch), we mathematically show that, given a pinch of length $2L$, the maximum displacement $\Delta z_{\max}$ a suspended particle may be advected by an individual pinch is

$$\Delta z_{\max} = \frac{8}{3}L. \tag{1}$$

Using the experimentally measured average value of the pinch length $2L = 0.14$ μm (*Holcman et al., 2018*), a typical pinch can then propel a particle by a maximum distance of $\Delta z_{\max} \approx 0.19$ μm.

This may, equivalently, be framed in terms of velocities. A transported particle experiences an average velocity during the contraction of at most $V_{\max} = 8L/3T$, where $T$ is the duration of a contraction or a relaxation. Using the pinch length as above and the average values of $2T = 0.213$ s (*Holcman et al., 2018*) this leads to the estimate $V_{\max} \approx 3.9$ μm/s, which is an order of magnitude smaller than the measured typical edge traversal speed of $\sim 45$ μm/s, consistent with the order-of-magnitude difference between the measurements and the predictions of the computational model.

Note that a slip boundary condition keeps the volume flux expelled out of (or driven into) the pinching regions unchanged, but decreases the maximum flow speed (see *Figure 3*). Therefore, a non-zero slip length would only decrease the theoretical maximum advective displacement $\Delta z_{\max}$ a suspended particle can achieve; therefore slip length plays no significant role in our theoretical argument to rationalise the weak pinching-induced transport.

## Marginal addition of coordinated contractility to luminal transport

The theoretical upper bound in the previous section for the maximum luminal transport producible by an individual pinch is realisable only in the hypothetical situation where the flow generated by the pinch is all directed to one end of the tubule that is when the other end is effectively blocked. Moreover, content transport produced during the contraction of a single tubule would then be reversed when the tubule relaxes back to its original state, with a single pinch site expected to exhibit only reciprocal (i.e. time-reversible) motions. Any advection contributing to edge

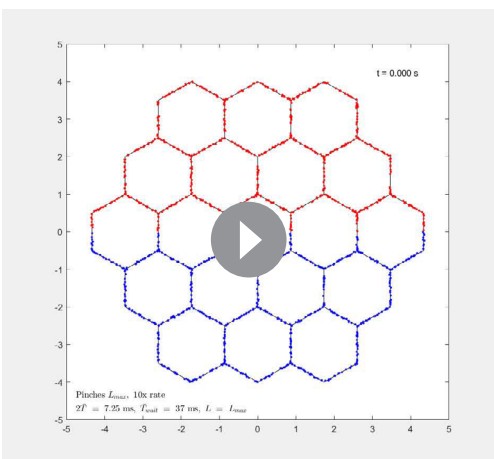

**Video 5.** Mixing in time in an active honeycomb network pinching with maximally long pinches at 10 times the original rates.
https://elifesciences.org/articles/93518/figures#video5

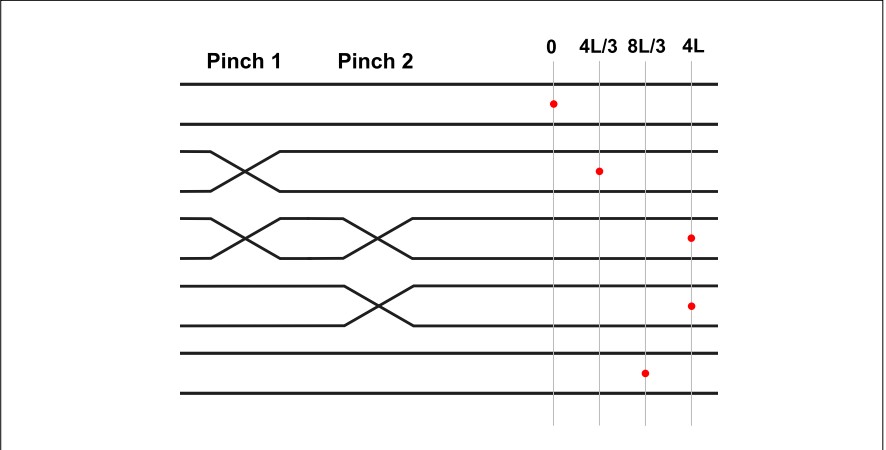

**Figure 5.** Illustration of coordination mechanism allowing the interactions between two pinch in series to induce the net transport of a suspended particle; the mechanism is akin to small-scale peristalsis. Red dot indicates the position of a particle on the tubule's centreline at each step of the coordination mechanism.

traversals must thus be dominated by non-reciprocal motions of multiple pinches resulting in net displacements of solute particles.

The simplest system capable of producing non-reciprocal motions consists of two pinches, and the optimal sequence of motions to maximise the resulting advective particle displacement is illustrated in *Figure 5*. We show in Materials and methods (Extension to nonlinear interactions between two pinches) that this is indeed the optimal two-pinch coordination, which results in a time-averaged displacement equal to the upper bound derived in *Equation 1*. Since this optimal sequence of motions involves one pinch site starting a pinch halfway through the pinching of the other site, it is reasonable to estimate its duration as $3T$, and therefore an average particle speed of $8L/9T \approx 1.3$ µm/s. The low particle speed achievable by the optimal coordination between two pinches suggests that a very high level of coordination among multiple tens of pinches per tubule would be required to reproduce the measured edge traversals.

## A combination of high frequency and pinch length is required to replicate experimental particle speeds

Since the magnitude of ER contractility suggested by microscopy (*Holcman et al., 2018*) (pinch lengths and frequency) does not explain the measured speeds, we set out to explore different sets of parameters that may generate particle velocities closer to the experimental measurements. The currently achievable imaging spatiotemporal resolution limits the detection of tubular diameter contractility by microscopy. Therefore, it is reasonable to postulate that the relevant parameters may have been underestimated. We simulate ER transport varying individually or in combination the values of pinch duration $2T$, waiting time $T_{wait}$ between successive pinches on a tubule, and pinch length $2L$.

We first decrease both the original (*Holcman et al., 2018*) values of $T$ and $T_{wait}$ by the same factor of $1/\alpha$, where $\alpha \geq 1$, and simulate particle transport in the honeycomb network (*Figure 6a*). In effect, this simply 'fast-forwards' the flows in the original active network by a factor of $\alpha$, and Brownian particles of the original diffusivity are released into this sped-up flow. Instantaneous and edge traversal speeds exhibit corresponding increases when we increase the value of $\alpha$ (*Figure 6a*). An extreme value of $\alpha = 100$ produces an average edge traversal speed distribution that peaks at around 8 µm/s (*Figure 6a*). Similar results are observed in the C0 network from a COS-7 cell (*Figure 6b*). The longer tails of these distributions (compared to those from the honeycomb network) result from the variation in edge lengths in the real network, with shorter edges, across which edge traversals are correspondingly fast, contributing to the tails.

These results suggest that, in order to produce average edge traversal speeds of the same order as the experimental values, we would need an active network sped up by an unrealistic factor considerably greater than 100, probably on the order of $\alpha \sim \mathcal{O}(10^3)$ or higher and corresponding to multiple thousands of pinches occurring on average per second on each tubule. Similarly, it takes an extreme

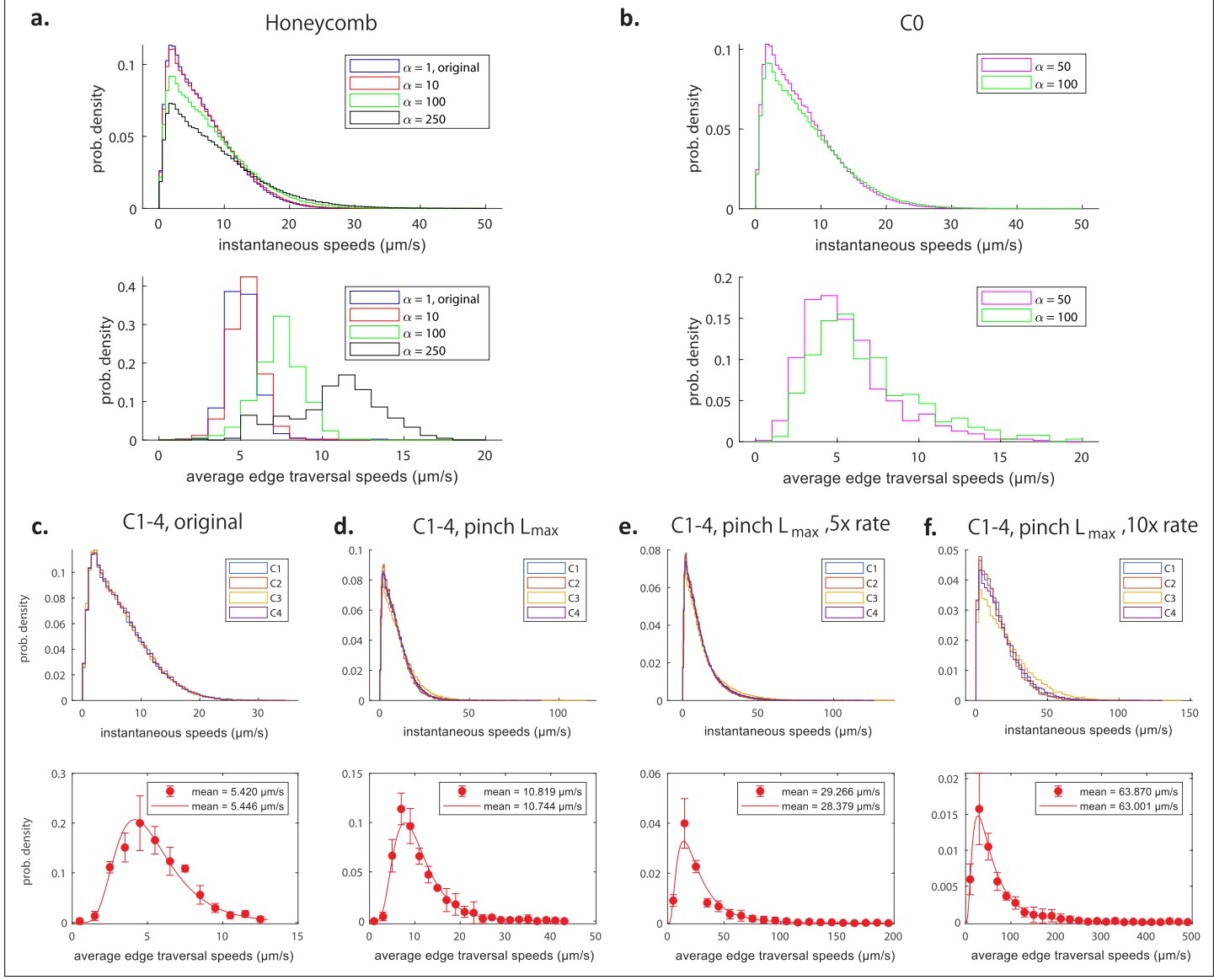

**Figure 6.** Impact of spatiotemporal pinch parameters on transport. Histograms of instantaneous speeds (top) and average edge traversal speeds (bottom), for an active honeycomb network (**a**) and the reconstructed C0 network from *Figure 9a–c* (**b**) with pinch parameters $\overline{T}$ and $\overline{T_{wait}}$ decreased to $1/\alpha$ times the original values from *Holcman et al., 2018*, and the same measured diffusivity $D = 0.6\,\mu\text{m}^2\text{s}^{-1}$. Histograms of instantaneous speeds (top) and average edge traversal speeds (bottom) for the C1–C4 networks from *Figure 9d* with varying pinch parameters: original parameters from *Holcman et al., 2018* (**c**); pinch length increased to the total length of the tubule (**d**); a fivefold increase in the rate of pinching and pinch length set to the total length of the tubule (**e**); a tenfold increase in the rate of pinching and pinch length set to the total length of the tubule (**f**). Bottom rows: points indicate mean ± 1 SD over the four networks (C1–C4) of normalised frequencies in each speed range; curves are log-normal distributions fitted to all average edge traversal speeds for each set of pinch parameters; insets show means of original simulation results and of fitted distributions.

increase in pinch site size spanning the entire length of an average tubule, only to yield an average edge traversal speed of $\sim 10$ µm/s (*Figure 6d*).

Next, we attempt to obtain a better fit to experimental data by combining changes in both the pinches' time and geometry parameters. In *Figure 6e, f*, these maximally long pinches are sped up by a factor of $\alpha = 5$ and 10, respectively, which yields speeds averaging around 30 µm/s and above 60 µm/s, respectively. Notably, the tail of the speeds distribution appear longer than that seen in the experiments.

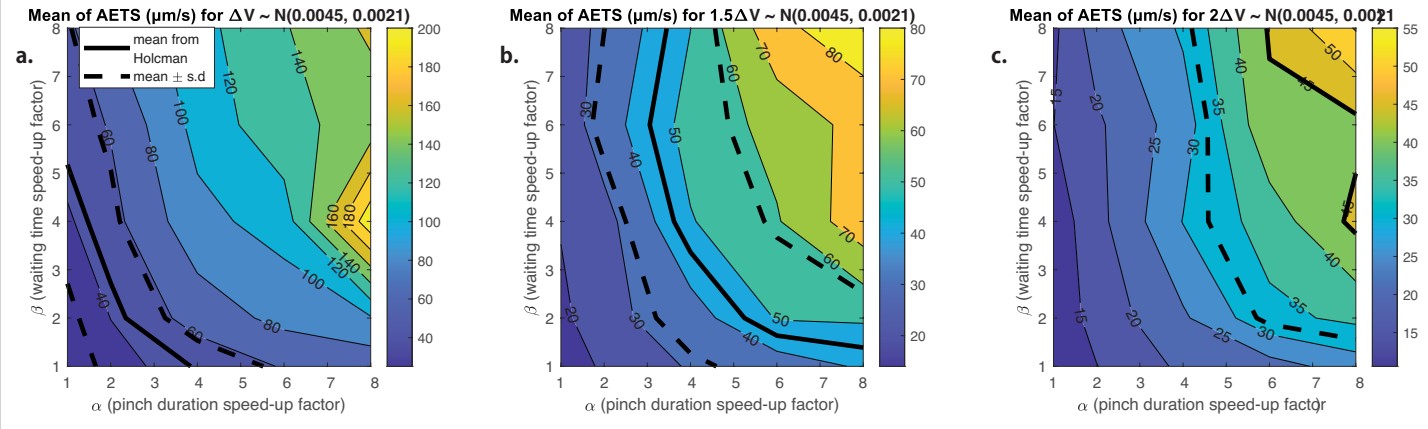

**Figure 7.** Transport driven by contracting tubular junctions. Contour plots of the mean values of the average edge traversal speeds obtained from simulations of our model in a junction- and tubule-driven C1 network from *Figure 9* with different values of $(\alpha, \beta)$ and with contraction volumes $\Delta V$ expelled during each contraction drawn from (**a**) the normal distribution estimated for the junction volumes N(0.0045, 0.0021) (in µm³); (**b**) two-thirds the estimated normal distribution for the junction volume; and (**c**) half the estimated distribution for the junction volume. Thick solid black lines indicate the mean of the average edge traversal speed distribution reported in *Holcman et al., 2018* (45.01 µm/s) and thick dotted black lines indicate mean ± SD (45.01 ± 12.75 µm/s).

## Luminal transport kinetics derived from contractile ER tubular junctions and sheets

As shown above, establishing effective transport in a tubular constriction-driven model based on realistic ER fluid dynamics requires a set of questionable assumptions, compelling us to explore alternatives. Thus, we set out to investigate how ER luminal transport would be impacted by the contractility of its structural components with volumes larger than tubules: (1) the tubular junctions (*Figure 1c*), (2) the perinuclear ER sheets (*Figure 1f*), and (3) the peripheral sheets (*Figure 1d*).

First, we run numerical simulations of transport driven by contracting junctions on an ER network (C1 from Figure 9d; sketch of junctions in *Figure 1c*). Since junctions contracting at the tubular pinch temporal parameters measured experimentally yielded inadequate transport, we consider contractions/relaxations with duration $2T$ exponentially distributed with a mean of $\alpha^{-1}$ times the original value in *Holcman et al., 2018* and the waiting time $T_{wait}$ between subsequent contractions/relaxations exponentially distributed with a mean $\beta^{-1}$ times the original value (i.e. values of $\alpha > 1$ and $\beta > 1$ reflecting faster and more frequent pinches). Naturally, the results depend on the choice of the volume $\Delta V$ expelled by a junction during each contraction. Using fluorescence microscopy images, we may estimate the volume in junctions (see Materials and methods, Experimental estimates of junction volumes). We assume that the junction volumes are drawn from a normal distribution with the same mean and SD as our data set that is the distribution $N(0.0045 \text{ µm}^3, 0.0021 \text{ µm}^3)$. Volumes greater than the maximum value in our experimental estimate (0.0081 µm³) or less than the minimum value (0.0020 µm³) are rejected.

We show the results in *Figure 7a*. The thick solid line illustrates a set of values of $(\alpha, \beta)$ which produces distributions of average edge traversal speeds with means similar to the experimental values in *Holcman et al., 2018*, and the dashed lines 1 SD away. We are interested in values of $(\alpha, \beta)$ close to unity because this corresponds to junctions which pinch with similar pinch durations and frequencies as the experimentally observed tubule pinches and are therefore biologically plausible; $(\alpha, \beta) \approx (2.5, 2)$ is the closest pair on this line to unity. However, only for $\alpha \approx 1$ do we obtain average edge traversal speed distributions of reasonable shapes (i.e. approximately Gaussian), which would require a large $\beta = 5$ to match experimental results quantitatively.

Furthermore, the assumption that the entire junction volume is expelled during a contraction may not be realistic, as it would require extreme bending/extension of tubule walls and the ability of the entire junction to empty and fill out during each contraction. Reducing the volume $\Delta V$ expelled in each contraction to two-thirds of the estimated distribution of the junction's volume (results shown in *Figure 7b*) or to half (*Figure 7c*) causes the average edge traversal speeds to drop considerably.

The lower the proportion of the junction volume expelled during a contraction, the faster the pinches are required to be (i.e. large value of $\alpha$) and the larger the frequencies of the pinch events (large $\beta$) in order to produce reasonably high average edge traversal speeds.

Next, we consider the particle transport generated by two types of ER sheets contracting and relaxing over a duration $2T$ (see Materials and methods for details). The perinuclear sheets are shaped as contiguous layers of flat cisternae with a luminal volume larger than the tubules branching from these structures (see sketch in *Figure 1f*). Accordingly, their contraction with $2T = 5$ s and $V_{sheet} = 10$ μm³ yields a mean average traversal speed of 35 μm/s consistent with the single-particle tracking experiments (see *Figure 8b*). However, the speeds distribution tails towards higher values (*Figure 8a, b*), something that has not be observed experimentally so far; it may be that high velocities cannot be recovered in experiments due to the constraints on linkage distance imposed to ensure trajectory fidelity in particle tracking.

Furthermore, our simulations reveal that the flow decays sharply with distance away from the sheet where it originated as it branches out into tubules. This is illustrated in *Figure 8c, d* and we further quantify the spatial gradient of the average edge traversal speeds across Cartesian 2D coordinates in *Figure 8e*, revealing the stark contrast between the homogeneous profile for contracting junctions (red dotted line) vs sheet-driven transport (blue solid line). The short range of influence afforded by contracting perinuclear ER sheets thus argues against its ability to sustain mixing flows and fast particle transport on the distal tubular network.

Instead, we explore whether the peripheral sheets (i.e. the smaller flat inter-tubular ER regions, see sketch in *Figure 1d*) may overcome the range limit. The peripheral sheets have volumes significantly larger than tubules and junctions, which we estimated at 0.12 μm ± 0.04 μm³ (see Figure 13 and Materials and methods, Experimental estimates of volumes of peripheral sheets for details). This suggests that the peripheral sheets could produce average edge traversal speeds compatible with experimental measurements. Assuming that a sheet typically occupies the area enclosed by a 'triangle' of tubules, we may incorporate the transport inside a sheet-driven network using our model for contracting junctions (also referred to as 'nodes' in our graph theoretical methodology), but with junction volumes set to the measured sheet volumes (see Experimental estimates of volumes of peripheral sheets for details) and with either (1) each node expelling one third of the volume expelled by a contracting sheet (since a peripheral sheet is in contact with three nodes on average), or (2) a contracting node expelling the entire volume expelled by a contracting sheet, but with only one-third of the junctions actively contracting at any one time. If we further assume that during each contraction a peripheral sheet expels half its total volume so that in simulation (1) node $k$ expels a volume $V_k/6$ in each contraction and in simulation (2) each active node $k$ expels a volume $V_k/2$, then nodes which contract at rates $\alpha^{-1} = 2.5$ and $\alpha^{-1} = 5$ times slower than the tubule pinches in *Holcman et al., 2018* in simulations (1) and (2), respectively, give average edge traversal speeds in the correct range of 40 μm/s (*Figure 8f, g*). The contraction of peripheral sheets may thus be offered as a plausible mechanism for fast luminal transport, provided they are able to contract with sufficiently high rates.

Furthermore, by relating the work done by a peripheral sheet contraction to the dissipation due to the flows induced inside the sheet and in the rest of the network, the energetic cost of one such contraction may be estimated to be of the order of 1000 molecules of ATP (see Materials and methods, Energetic cost estimate for contracting peripheral sheets for details). For reference, a kinesin motor protein uses one molecule of ATP to move 8 nm (*Coy et al., 1999*; *Schnitzer and Block, 1997*), whereas a muscle fibre consumes hundreds of thousands of ATP per second (*Barclay, 2017*). These peripheral sheet contractions may be directly ATP-driven, or they might come as a result of other mechanical processes in the cell, similar to mechanisms generating cytoplasmic mixing flow (*Klughammer et al., 2018*; *Guo et al., 2014*; *Lin et al., 2016*).

## Discussion

A better understanding of the interplay between ER structure and function via luminal transport speeds may hold clues to explaining the sensitivity of cells with extensive projections, such as sensory and motor neurons, to defects in ER shaping proteins. It is tempting to speculate that disturbed ER luminal transport, the kinetics of which is particularly important for communication across vast axonal lengths, underlies selective vulnerability of long neurons.

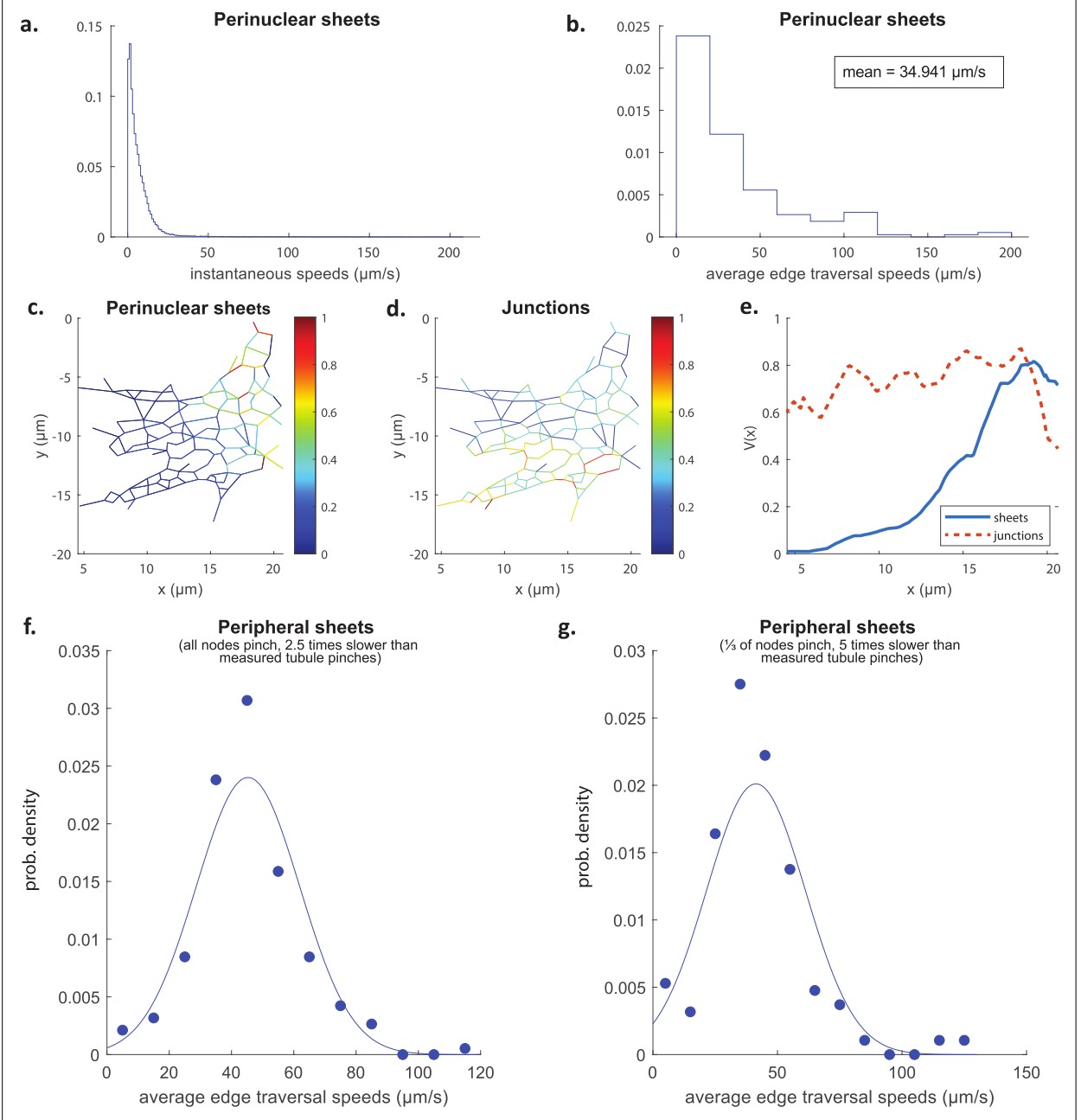

**Figure 8.** Transport driven by contracting perinuclear sheets or peripheral sheets. Distributions of instantaneous speeds (**a**) and average edge traversal speeds (**b**) obtained from simulations of the C1 network from **Figure 9** driven by the contraction of a perinuclear sheet. In these simulations, the sheet undergoes one contraction + relaxation lasting $2T = 5$ s, and expels a volume $V_{sheet} = 10$ μm³ of fluid during a contraction. Colour maps of normalised average edge traversal speeds obtained from simulations of the C1 network from **Figure 9** driven by contraction of tubules + sheets (**c**) and junctions + tubules (**d**), respectively. (**e**) The speeds averaged along the $y$ direction of the network, $V(x)$, are plotted against $x$, to effectively project the information onto one dimension from c (blue solid line) and d (red dashed line). (**f, g**) Histograms of average edge traversal speeds (dots) and normal fits (lines) and mean AETS (inset) in C1 networks with parameters adjusted as follows to approximate a network with peripheral sheets: Junction $k$ expels a volume $V_k/6$ of fluid in each pinch, the pinches are $\alpha^{-1} = 2.5$ times slower than the original tubule pinches, and all nodes actively pinch (**f**); and node $k$ expels a volume $V_k/2$ of fluid in each pinch, the pinches are $\alpha^{-1} = 5$ times slower than the original tubule pinches, and only a third of the nodes actively pinch (**g**).

The motion of solutes in cellular compartments is now understood to be facilitated by active components. This is evident from direct motion measurements and the dependence of motion speed on the availability of ATP-contained energy (*Dayel et al., 1999*; *Nehls et al., 2000*; *Holcman et al., 2018*; *Koslover et al., 2017*). The origin of these active driving forces is, however, challenging to identify. The cytoplasm's currents are often believed to originate from the motion of large complexes such as ribosomes and large vesicle cargo shuttled by cytoskeleton-mediated motorised transport (*Koslover et al., 2017*). In the case of enhanced transport in the lumen of the ER tubules, the contractility of tubules has been suggested as the flow generating mechanism, and indeed such tubule deformations have been observed in microscopy (*Holcman et al., 2018*). However, establishing a direct empirical link between tubule contraction and active flows, or experimentally testing other hypotheses for the driving mechanism behind ER solute transport, remain currently unattainable. In this study, we thus propose a physical modelling approach, which provides a platform to explore the nanofluidics behaviour of biological systems such as the ER network. The outcomes of our simulations for a contractile ER argue against the plausibility of local pinch-driven flow; pinches with frequency and size on the order of those estimated by microscopy yield significantly lower speeds than single-particle tracking measurements, as well as no enhancement of mixing beyond that from passive diffusion. The deficit stems from the fact that the displaced fluid volume upon local contraction is too small to generate sufficient particle transport.

Given the uncertainty of the empirical measurements for ER tubule deformation, due to the limits in the spatiotemporal resolution of organelle structure imaging, the pinch parameters may have been significantly underestimated. This sanctions exploring a wider range of spatiotemporal parameters in our simulations, which revealed that a combination of a higher frequency with a much larger pinch length may provide higher particle speeds that are comparable to the single-particle tracking measurements. Furthermore, our modelling results suggest the possibility of transport by luminal width contractions of larger volume ER subdomains (which are contiguously interconnected with the network). In that respect, the contraction of peripheral (i.e. inter-tubular) sheets, in particular yields speed values in a plausible range provided they are able to contrast fast enough. In contrast, the contraction of large-volume perinuclear sheets leads to fast transport but with a limited spatial range and with flows that would not impact transport beyond a few microns into the peripheral network. Similarly, the alternative scenario of contractility of tubular junctions appears unlikely as particle speeds similar to experiments could only appear for junctions pinching at up to eight times faster than experimentally observed contracting tubules.

The modelling approach in this study, although focused on the ER network, provides a step forward towards understanding intraorganellar fluid dynamics. Our simulation results rule out several scenarios, which seemed physically intuitive but are nevertheless unable to explain the observed enhanced luminal transport. Furthermore, our results generate a set of potentially testable predictions that can be used to validate or refute each of the envisaged transport mechanisms. For example, as fluid expulsion from peripheral sheets appears to be in broad in agreement with single-particle tracking measurements data under even conservative assumptions, future measurements may explore whether an active luminal motion is more pronounced and faster in proximity to peripheral sheets. Moreover, a set of improved spatiotemporal resolution measurements of particle tracking and structural contractions will be needed to complete the physical picture of ER luminal transport.

While our study allowed us to test different plausible scenarios, questions remain open as to the force generation mechanisms responsible for the observed pinching dynamics. We have estimated in Estimate of forces required for pinches that forces on the order of 30 pN would be required to periodically contract the tubules as seen empirically. The force exerted by a single molecular motor has been estimated to be on the order of 6 pN (*Fisher and Kolomeisky, 1999*), so the required forces for pinching may be provided by several motors working together (*Jülicher et al., 1997*). Recently discovered hydro-osmotic instabilities could also contribute to shape fluctuations and pinching, although they are predicted to have much longer wavelengths than the typical size of a pinching region (*AlIzzi et al., 2018*). In contrast, the mechanisms involving topological remodelling of the ER, such as the well-documented process of ring closure (*Guo et al., 2018*), occur over timescales of minutes and therefore cannot account for the millisecond-scale transport measured here.

We have used in silico fluidics modelling for our conclusions, particularly in identifying new physically permitted mechanisms of luminal propulsion, and our simulations should be regarded as theoretical

predictions demonstrating physical plausibility rather than mere speculation. We have explored theoretically the consequences of structural fluctuations which presumably do take place. Fluctuations in width of the flat ER areas, in particular, are expected since the structures appear dynamic in live light microscopy and narrowing/extension points are observable in electron microscopy (*Heinrich et al., 2021*; *Wu et al., 2017*). Furthermore, variability in the areas of flat ER that are observable in electron microscopy can be explained by capturing the structures in different states of fluctuations. It should be expected that the mobile elastic structure such as ER sheets would not exhibit rigidity required to prohibit fluctuations.

In conclusion, it is worth emphasising that our in silico fluid dynamical modelling reveals that for structural fluctuation-based mechanisms to facilitate luminal motion, assumptions currently not supported by empirical data are required. This warrants explorations of alternative mechanisms for ER luminal transport, for example anomalous diffusion driven by the fluctuations of macromolecular complexes (*Koslover et al., 2017*) or by osmotic forces, as previously suggested (*Holcman et al., 2018*).

## Materials and methods

Here, we describe the mathematical and physical model for the fluid mechanics and transport driven by active contractions in the ER. This requires the introduction of a network model for the geometry of the ER (Network modelling) and individual pinches (Pinch modelling) and a framework for the hydrodynamics of pinching tubules (Hydrodynamic modelling). Our solution method for the flows inside our network is then described in Solving the hydrodynamic network model. From simulations of Brownian particles advected by these flows (Simulating particle transport) quantitative measures of particle transport (Data processing: instantaneous speeds and average edge traversal speeds) are extracted for comparison with experiment. Boundary slip is incorporated into our model in Incorporating slip boundary conditions. The force required to pinch a tubule is estimated in Estimate of forces required for pinches. In Derivation of theoretical bounds for active flows driven by pinching tubules, we present in detail the derivations of the theoretical results discussed in Theoretical analysis of advection due to a single pinch explains weak pinching-induced transport (transport upper bound by a single pinch) and Marginal addition of coordinated contractility to luminal transport (coordination of pinches). Finally, modifications of our model to explore alternative flow generation mechanisms are discussed in Modelling of alternative flow generation mechanisms, and the energetic cost in contracting a peripheral sheet is estimated in Energetic cost estimate for contracting peripheral sheets.

### Network modelling

We represent the 'skeleton' of a two-dimensional ER network as a planar graph with each node assigned a position $\mathbf{x} \in \mathbb{R}^2$. Given an edge of the network labelled $(i,j)$ and of length $|\mathbf{x}_i - \mathbf{x}_j| = l$, we model the lumen of the corresponding tubule to occupy a cylinder of radius $R$ whose axis lies along the edge and has length $l$. This assumption avoids the intrinsic difficulty in defining a precise boundary between a tubule and a tubular junction, as well as leading to a simplified model of the intra-nodal dynamics of a solute particle (see below); since the tubules are long compared to the size of the junctions, the impact of their small overlap can be safely neglected.

We mathematically reconstruct a model ER network, which we refer to as C0, using the skeleton image of the COS7 ER network given in Supplementary file 3 of *Holcman et al., 2018*; this network is reproduced in *Figure 9a*. We use the multi-point tool in ImageJ (*Schneider et al., 2012*) to place numbered points at the positions of the nodes on the source skeleton image, and then obtain a list of the indices and position coordinates of each node. The edges are then manually tabulated as a list of pairs of nodes; this gives us all the information required to construct a mathematical graph as shown in *Figure 9b*, with the original network superimposed on the mathematical model in *Figure 9c*. Note that in what follows we work with the largest connected component of the source skeleton image in order to study transport in a fully connected network. We also use the same procedure to extract the graph structures from microscopy images, of four smaller ER networks which we label C1–C4 (original network with mathematical graph superimposed in *Figure 9d*). In *Figure 9e*, we show the distributions of the edge (tubule) lengths in each of the C0–C4 networks, as well as the mean edge lengths, the mean degrees (the degree of a node is the number of edges connected to it), and the number of

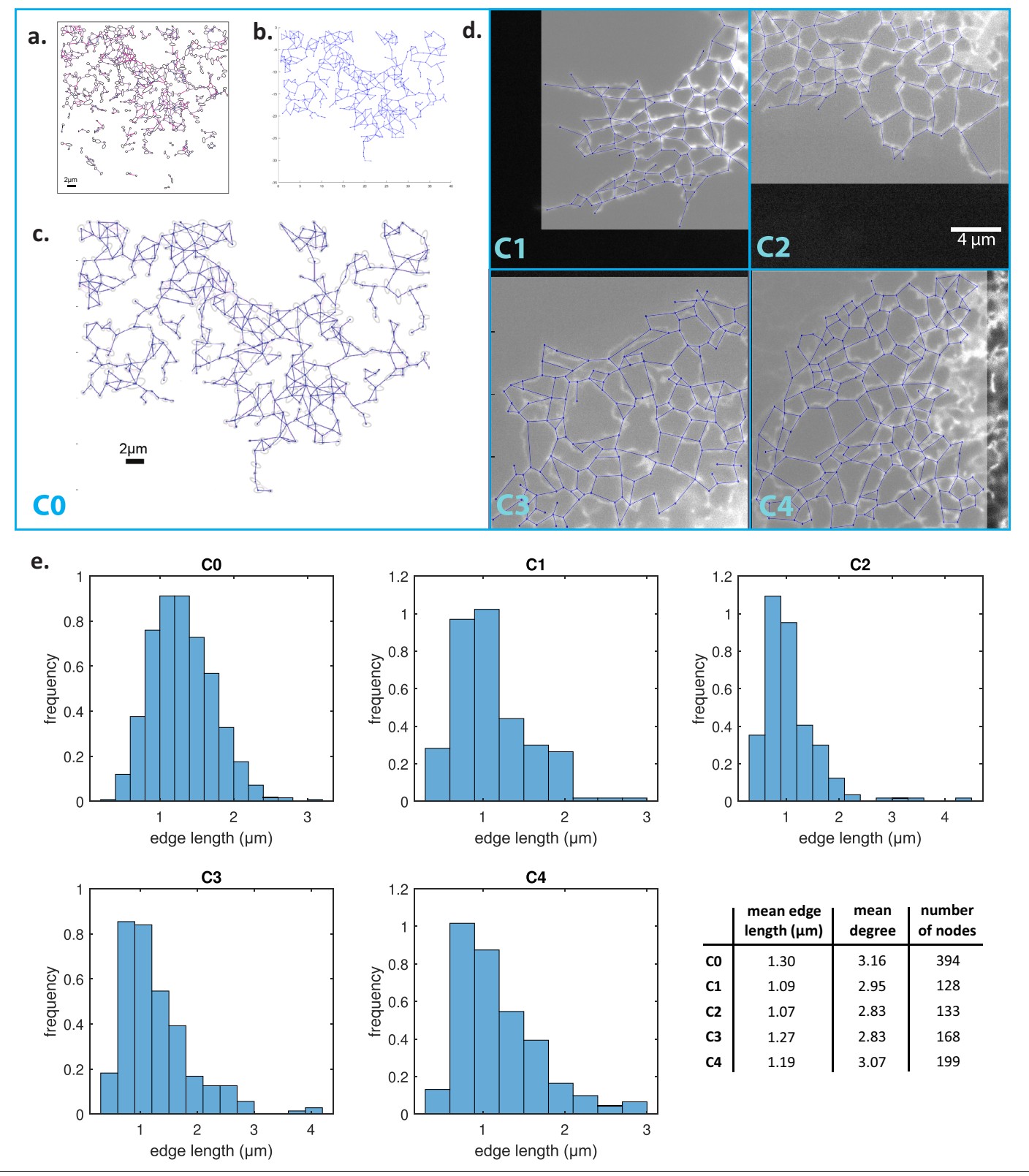

**Figure 9.** Model ER networks and their statistics. (**a**) Skeleton image of COS7 ER reproduced from Supplementary figure 3 of *Holcman et al., 2018*. (**b**) Model ER graph (blue solid lines) reconstructed from (**a**) using ImageJ. (**c**) Experimental images in (**a**) superimposed with mathematical model from (**b**). (**d**) Microscopy images of four different COS7 ER networks (labelled C1–C4) with reconstructed model networks (blue solid lines) superimposed. (**e**) Distributions of edge lengths in the C0–C4 networks. Bottom right: mean edge lengths, mean degrees (i.e. number of edges connected to a node) and number of nodes of the C0–C4 networks.

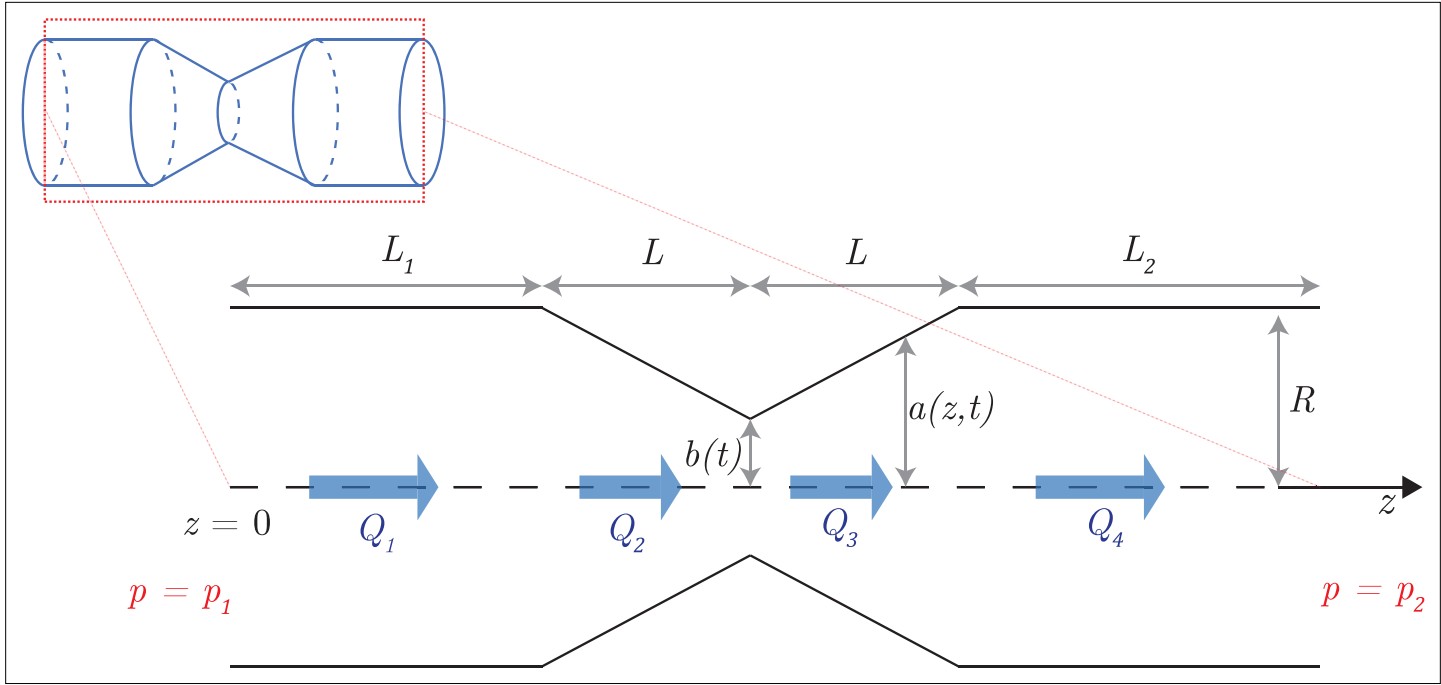

**Figure 10.** Mathematical model of a pinching tubule. The tubule has a radius of $R$ outside the pinch, $a(z,t)$ in the pinch (where $z$ is the axial coordinate), and $b(t)$ at its narrowest point that is the centre of the pinch. The portion of the tubule before the tubule has length $L_1$ while that after has length $L_2$; the pinch is symmetric and has a length $2L$. $Q_1, Q_2, Q_3$, and $Q_4$ denote the volume fluxes through the tubule in the four different regions as indicated. The pressures at the end of the tubule are $p = p_1$ at $z = 0$ and $p = p_2$ at $z = L_1 + 2L + L_2$.

nodes. The mean edge lengths are around 1 µm and mean degrees are approximately 3. In order to compare the biological network to an idealised ER system, we also consider a honeycomb network, that is one where every node (apart from those at the boundaries) has a degree of 3, with all edge lengths exactly 1 µm.

## Pinch modelling

To describe the pinches (*Figure 1e*), we consider a model where the kinematics of each pinch is fully prescribed in time. We therefore assume that the active biochemical forces responsible for the deformation of the tubules balance with the elastic resistance of the tubules and with the dissipative forces in the fluid in such a way that the pinches occur as described.

The geometrical model for a pinch is illustrated in *Figure 10*. Each tubule is assumed to have a pinch site at its midpoint (i.e. with $L_1 = L_2$); we make this simplifying assumption in our simulations, after verifying that more general pinch locations ($L_1 \neq L_2$) have virtually no effect on particle transport. The pinching events occur at the pinch sites stochastically and independently of each other. Each pinch is defined geometrically by three parameters from a random distribution (see below): (1) the duration of a pinch $2T$, (2) the time $T_{wait}$ between the end of a pinch and the beginning of a new one on the same site, and (3) the length of a pinch $2L$ (see *Figure 10*). We assume for simplicity that all pinches are axisymmetric so that using the notation of *Figure 10*, each tubule remains a cylinder of time-varying radius denoted by $r = a(z,t)$. Pinches are assumed to have reflectional symmetry about the plane in the cross-section through the centre of the pinch (i.e. in *Figure 10* each pinch is characterised by the same length $L$ on either side of it). The total length of a tubule is denoted by $l = L_1 + 2L + L_2$, so the portion before the pinch has length $L_1$, and that after the pinch is of length $L_2$.

We model the geometrical profile of each pinch of length $2L$ as following a linear radius change (see *Figure 10*). Within a pinch located at $z = z_0$, the radius of the cylinder is therefore given by

$$a(z,t) = b(t) + [R - b(t)]\frac{|z - z_0|}{L}, \quad -L \leq z - z_0 \leq L. \tag{2}$$

The time-varying function $b(t) \leq R$ is therefore the minimum pinch radius in the centre of the tubule. As the simplest modelling choice, we assume in our simulations that $b$ changes in time sinusoidally and thus will use $b(t) = (R + b_0)/2 + (R - b_0)\cos(\pi t/T)/2$, where $t$ is time after the pinch begins and $b_0$ is the value of $b$ halfway in time through the pinch. Choosing a smooth time variation for the function $b(t)$ will ensure the continuity of fluxes in time (see *Equation 10* later). We have verified that changing the pinch shape to a smoother geometry has essentially no impact on the fluid volume expelled/taken in, and therefore no significant effect on the flows/transport.

In our simulations, the stochastic pinch parameters are drawn from the distributions measured in *Holcman et al., 2018*. The pinch duration $2T$ is therefore drawn from an exponential distribution with rate parameter $\lambda = \ln 10/0.167$ s$^{-1}$ (so the mean value is $\overline{2T} = 0.0725$ s), the time between pinches $T_{wait}$ from an exponential distribution with rate parameter $\lambda_{wait} = \ln 10/0.851$ s$^{-1}$ (mean value $\overline{T_{wait}} = 0.370$ s) and the pinch length $2L$ from a uniform distribution with mean μm and variance $\sigma^2 = (0.06^2/\ln 10)\,\mu m^2 \approx (0.040\ \mu m)^2$. Throughout we set the tubule radius $R = 30$ nm and $b_0 = 0.01R$ (recall from *Figure 10*).

## Hydrodynamic modelling

### Hydrodynamics of network

We assume that the fluid occupying the ER network is Newtonian. Flow inside the network occurs at low Reynolds number, which can be justified as follows. From experiments in *Holcman et al., 2018*, we know that velocity scale relevant to ER flows is of the order $U \sim 10^{-5}$ ms$^{-1}$. With a typical tubule radius $R \sim 10^{-8}$ m and a kinematic viscosity of at least that of water $\nu \approx 10^{-6}$ m$^2$ s$^{-1}$, we obtain a Reynolds number of the order $\mathrm{Re} = UR/\nu \sim 10^{-7}$, so the flow is indeed Stokesian and inertial effects in the fluid can be safely neglected.

At a typical instance in time, a tubule undergoing contraction (or relaxation) causes a net volume flux to exit (or enter) the corresponding tubule. In the context of our graph theoretical model, we therefore model each pinch site as a 'pinch node' generating a net hydrodynamic source/sink whenever the tubule contracts/relaxes.

However, since it is not guaranteed that the total volume created by all pinching events always adds up to zero, we need a mechanism for the corresponding net volume to exit, or enter, the network. We achieve this through a number of 'exit nodes' that allow mass to be globally conserved, located at the periphery of the network. From a hydrodynamic standpoint, we impose the pressure condition $p = p_0$ at each exit node to model their connection to a large fluid reservoir.

We numerically tested the robustness of our results to the details of the exit nodes by repeating simulations with different configurations. The exact choices of exit nodes turn out to not affect the transport results shown below provided there are sufficiently many of them to avoid channelling the entire network's worth of pinch-induced flow into a few tubules towards the exterior, thereby producing artificially strong flows.

### Hydrodynamic model for a pinch

#### Flow rate

The velocity field in a straight cylindrical tubule at low Reynolds number is the classical parabolic Poiseuille flow (*Batchelor, 1967*). When integrated over a cross-section of the tubule, this flow yields the Hagen–Poiseuille law relating the pressure change $\Delta p$ across a length $l$ of a tubule to the a net volume flux $Q$ in the positive axial direction,

$$\Delta p = -\frac{8\mu l}{\pi R^4}Q, \tag{3}$$

where $\mu$ is the dynamic viscosity of the Newtonian fluid. Note that when $Q > 0$ our notation leads to $\Delta p < 0$, meaning that the pressure decreases across the length of the channel.

The result in *Equation 3* is valid for a straight (i.e. not pinching) tubule, and we need to generalise it to the case of a pinching tubule. Consider first a more general axisymmetric pipe whose radius $a(z, t)$ varies with axial position $z$ and time. We may use the long-wavelength (lubrication) solution to Stokes' equations for the streamwise velocity $u(z, r, t)$ and flux $Q(z, t)$ inside such a pipe into which a flux $Q_1(t)$ enters at $z = 0$ (*Alim et al., 2013*),

$$u(z, r, t) = 2 \frac{Q(z, t)}{\pi a(z, t)^2} \left[ 1 - \left( \frac{r}{a(z, t)} \right)^2 \right], \tag{4}$$

where

$$Q(z, t) = Q_1(t) - 2\pi \int_0^z a(\tilde{z}, t) \frac{\partial a(\tilde{z}, t)}{\partial t} d\tilde{z}. \tag{5}$$

The equality in *Equation 5* can be derived using an intuitive mass conservation argument, independently of the inspired ansatz in *Equation 4*. Conservation of mass inside a small section $[z, z + \delta z]$ of the cylinder requires $Q(z) - Q(z + \delta z) = 2\pi a \dot{a} \delta z + \mathcal{O}(\delta z^2)$. Considering the limit $\delta z \to 0$ and integrating the resulting expression for $\partial Q / \partial z$ from 0 to $z$ yields *Equation 5*.

In order for the no-slip boundary conditions at $r = a(z, t)$ to be satisfied, and also to satisfy the incompressibility condition $\nabla \cdot \mathbf{u} = 0$, the radial component of the velocity, $v(z, r, t)$, is necessarily non-zero and given by *Alim et al., 2013* as

$$v(z, r, t) = \frac{\partial a(z, t)}{\partial t} \frac{r}{a(z, t)} \left[ 2 - \left( \frac{r}{a(z, t)} \right)^2 \right] + 2 \frac{\partial a(z, t)}{\partial z} \frac{Q(z, t) r}{\pi a(z, t)^3} \left[ 1 - \left( \frac{r}{a(z, t)} \right)^2 \right]. \tag{6}$$

Our geometrical model for each pinch in Pinch modelling yields straightforwardly a piecewise linear expression for $a(z, t)$ in different regions of the tubule, with the time dependence entering only through the value of the pinch radius $b(t)$. Denoting by $Q_i$ ($1 \le i \le 4$) the fluxes in the four regions of the pinched tubules shown in *Figure 10*, we may substitute the linear shape functions into *Equation 5* and obtain

$$Q_2 = Q_1 - \frac{2\pi \dot{b} (z - L_1)^2}{L} \left( \frac{R}{2} - \frac{(R - b)(z - L_1)}{3L} \right), \tag{7}$$

$$
\begin{aligned}
Q_3 \quad &= Q_1 - 2\pi \dot{b} L \left( \frac{R}{6} + \frac{b}{3} \right) \\
&- 2\pi \dot{b} [z - (L_1 + L)] \left\{ b + \frac{R - 2b}{2L} [z - (L_1 + L)] - \frac{R - b}{3L^2} [z - (L_1 + L)]^2 \right\},
\end{aligned}
\tag{8}
$$

$$Q_4 = Q_1 - 2\pi \dot{b} L \left( \frac{R}{3} + \frac{2b}{3} \right). \tag{9}$$

Note that the final expression may be rearranged as

$$Q_4 - Q_1 = -2\pi \dot{b} L \left( \frac{R}{3} + \frac{2b}{3} \right), \tag{10}$$

which may be interpreted as the instantaneous volume source/sink during a contraction/relaxation at a pinch site.

## Pressure drop

We next need to compute the pressure drop in the pinches. We integrate the $z$-component of the Stokes equation

$$\frac{\partial p}{\partial z} = \frac{1}{r} \frac{\partial}{\partial r} \left( r \frac{\partial u}{\partial r} \right) + \mu \frac{\partial^2 u}{\partial z^2}, \tag{11}$$

along $0 \le z \le L_1 + 2L + L_2$, and use the solution for $u$, to obtain

$$
\begin{aligned}
\Delta p \equiv p_2 - p_1 \quad &= -\frac{8\mu}{\pi} \int_0^{L_1 + 2L + L_2} \frac{Q(z, t)}{a(z, t)^4} dz + \mu \frac{\partial u}{\partial z} \Big|_0^{L_1 + 2L + L_2} \\
&\approx \underbrace{-\frac{8\mu}{\pi} \int_0^{L_1 + L} \frac{Q(z)}{a(z, t)^4} dz}_{I_1} \underbrace{-\frac{8\mu}{\pi} \int_{L_1 + L}^{L_1 + 2L + L_2} \frac{Q(z)}{a(z, t)^4} dz}_{I_2}.
\end{aligned}
\tag{12}
$$

Here, the second term on the right-hand side of *Equation 12* has vanished because $\partial_z u \propto \partial_z(Q/a^2)$ but $Q$ and $a$ are approximately constant at the entrance and exits of the tubule when the pinch site is sufficiently far from the ends of the tubule so that the flow is fully developed there.

Using the expression (*Equation 7*) for $Q_2$, an integration yields

$$I_1 = \frac{-8\mu Q_1 L_1}{\pi R^4} - \frac{8\mu}{\pi} \left[ \frac{LQ_1}{3(R-b)} \left( \frac{1}{b^3} - \frac{1}{R^3} \right) + \int_{L_1}^{L_1+L} \frac{Q_2(z) - Q_1}{a(z,t)^4} \, dz \right] \tag{13}$$

and, by symmetry,

$$-I_2 = \frac{8\mu Q_4 L_2}{\pi R^4} - \frac{8\mu}{\pi} \left[ -\frac{LQ_4}{3(R-b)} \left( \frac{1}{b^3} - \frac{1}{R^3} \right) + \int_{L_1+2L+L_2}^{L_1+L} \frac{-Q_3(z) + Q_4}{a(z,t)^4} \, dz \right]. \tag{14}$$

Subtracting these two results (and noting the integrals cancel out by symmetry) we obtain the modified Hagen–Poiseuille expression for a pinching tubule as

$$\Delta p = -\frac{8\mu}{\pi} \left[ \frac{L_1}{R^4} + \frac{L}{3(R-b)} \left( \frac{1}{b^3} - \frac{1}{R^3} \right) \right] Q_1 - \frac{8\mu}{\pi} \left[ \frac{L_2}{R^4} + \frac{L}{3(R-b)} \left( \frac{1}{b^3} - \frac{1}{R^3} \right) \right] Q_4. \tag{15}$$

Note that this relationship is linear in each of $Q_1$ and $Q_4$, and is to be solved alongside *Equation 10* to relate the flow rates and pressure drops to the change in size of the pinches. Importantly, the classical Hagen–Poiseuille law is recovered as $b \to R$ since *Equation 14* becomes in that limit

$$\Delta p = -\frac{8\mu(L_1 + 2L + L_2)Q_1}{\pi R^4}, \tag{16}$$

which agrees with *Equation 3* when taking $l = L_1 + 2L + L_2$.

## Solving the hydrodynamic network model

The incorporation of pinches as 'dummy nodes' into the graph theoretical framework of Network modelling along with *Equations 10 and 14* for the necessary pinch-related quantities allow us to reduce the problem of determining the time-dependent flows in an active pinching network into the simpler problem of solving at each instant for the instantaneous fluxes inside a 'passive' network with newly added nodes, appropriate sources/sinks, and modified pressure drops. Note that since the flows at these sub-cellular scales are inertialess (i.e. Stokes flows), we are able to effectively decouple time from our problem and solve the problem in the quasi-steady limit.

For each edge (i.e. tubule) $(i,j)$ in the network, we define $Q_{ij}$ to be the flow rate from node $i$ to node $j$, with the sign convention that flow is from $i$ to $j$ if $Q_{ij} > 0$. For mathematical convenience, we define $Q_{ij} = 0$ in all cases where $(i,j)$ is not an edge in the graph. The goal is to solve for the values of the $Q_{ij}$'s corresponding to each edge.

After the incorporation of the dummy nodes, we denote by $N$ the number of nodes and $E$ the number of edges. We label the nodes such that $\{1, \ldots, M\}$ denotes the $M$ exit nodes. Let $q_i$ be the source or sink carried by the $i^{\text{th}}$ node (so that $q_i = 0$ if $i$ is a normal node, $q_i$ is as specified by the RHS of *Equation 10* if $i$ is a pinch node, and $q_i$ is a quantity to be determined if $i$ is an exit node). Our $M + E$ independent variables are therefore $\{q_i | i = 1, \ldots, M\}$ (that is, the sources/sinks carried by the exit nodes), and the $Q_{ij}$'s corresponding to each edge. To obtain their values, we employ the viscous hydraulic analogues of Kirchhoff's laws.

### Kirchhoff's first law (K1)

The first equation is that mass is conserved at each junction, that is, for each node $i$ we have

$$\sum_j Q_{ij} = q_i, \tag{17}$$

which gives us therefore $N$ equations. Note that these equations together imply global conservation of mass, $\sum_{i=1}^{N} q_i = 0$.

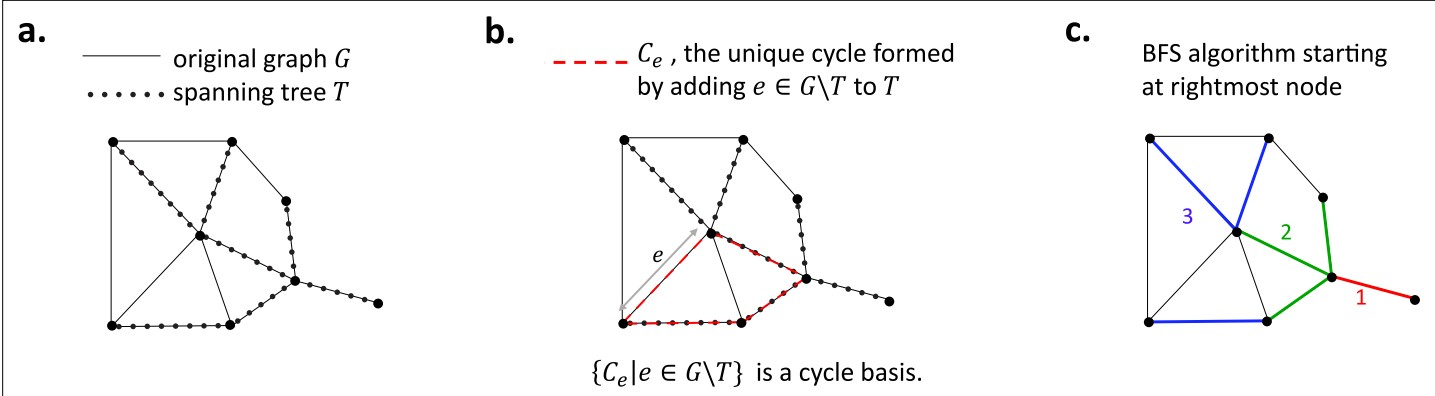

**Figure 11.** Elements of graph theory required to model the ER network. (**a**) A graph $G$ (black solid lines) and its spanning tree $T$ (black dots). (**b**) The unique cycle $C_e$ (red) formed by adding an edge $e \in G\backslash T$ to $T$. (**c**) A breadth-first search (BFS) starting at the rightmost node; the graph is explored in the order red, green, blue.

## Kirchhoff's second law (K2)

The second equation is a statement of consistency of pressure, namely that the pressure change around any cycle (i.e. closed loop) of the network is zero. Therefore, in a given cycle $C = \{v_1, v_2, \ldots, v_n, v_{n+1} = v_1\}$, if $\Delta p_{v_i v_{i+1}}$ denotes the pressure change from node $v_i$ to node $v_{i+1}$, we necessarily have

$$\sum_{i=1}^{n} \Delta p_{v_i v_{i+1}} = 0. \tag{18}$$

Note that the pressure change across a node is negligible.

The K2 statement in *Equation 18* applies to all cycles in the graph, which would give us more equations than we need since the vectors of coefficients in *Equation 18* are linearly dependent. Instead, we need a minimal set of linearly independent K2 equations, corresponding to the cycles in a cycle basis of the graph, and we need only apply K2 to these cycles.

To construct a cycle basis of the graph $G$, we use standard results from graph theory (*Wilson, 2015*). We first construct a spanning tree $T$, defined as a connected subgraph which contains all the nodes of $G$ and no cycle, as shown in *Figure 11a* on an example. Any tree $T$ has $N - 1$ edges. Therefore, there are $E - (N - 1)$ edges in the graph $G$ but not in the spanning tree $T$; for each such edge $e$, we denote by $C_e$ the unique cycle in the graph created by adding the edge $e$ to the tree $T$ (see *Figure 11b*). The set of all such cycles $C_e$ is then a cycle basis of $G$, and thus there are $E - (N - 1)$ cycles in this set. There are therefore $E - N + 1$ independent cycles in the cycle basis.

To compute the spanning tree $T$ we use a breadth-first search algorithm (*Goodrich and Tamassia, 2015*). We start from an arbitrary node and explore its neighbouring nodes. We add to $T$ any previously unexplored node (and the corresponding edge) which does not result in the creation of a cycle in $T$. We then repeat this (in an arbitrary order) on the neighbours of the previous generation of nodes added to $T$, until no more nodes are left to explore. This is illustrated on an example in *Figure 11c*.

A similar algorithm is also used to compute the cycles $C_e$ in the cycle basis. This time, however, denoting $e = (i, j)$, the algorithm is started from $i$ and set to terminate as soon as $j$ is visited, yielding a path in the tree from $i$ to $j$, which, together with the original edge $(i, j)$, completes a cycle.

## Pressure boundary conditions at exit nodes

At this point in the modelling, we have $M + E$ independent variables (the flow rates in each edge and at the exit nodes), $N$ equations from K1 (i.e. *Equation 17*), and $E - N + 1$ equations from K2 (i.e. *Equation 18*). The remaining $M - 1$ equations follow from requiring the exit nodes to be at the same pressure, modelling their connection to a common fluid reservoir. This is ensured by the $M - 1$ equations

$$\sum_{1 \to j} \Delta p = 0 \tag{19}$$

for $j = 2, \ldots, M$, where the sum $\sum_{1 \to j}$ is defined to be over any path from node 1 to node $j$. Note that thanks to Kirchhoff's second law, this quantity is independent of the specific path taken from 1 to $j$. We then solve the resulting linear system of $M + E$ equations numerically. Note that we do not need to specify the value of the fluid viscosity $\mu$ in our algorithm because it cancels out in the K2 equations, *Equation 18*, and in the pressure boundary condition equation, *Equation 19*.

## Simulating particle transport

With our solution for the flows in the active network at each instant of time, we now proceed to track the motion of Brownian particles inside the network using a discretisation of their stochastic equations of motion, as a model for the transport of proteins in the ER network.

We use the simplest approach where we superimpose Brownian motion onto advection by the flow inside each tubule. Let $\mathbf{x}(t)$ denote the position of a Brownian particle in a tubule and $\mathbf{x}_n$ the finite-difference approximation of $\mathbf{x}(n\Delta t)$, where $\Delta t$ is a discrete time step. The displacement of the particle at each time step can be obtained approximately using an explicit first-order Euler scheme

$$\mathbf{x}_{n+1} = \mathbf{x}_n + \mathbf{U}(\mathbf{x}_n, t)\Delta t + \mathbf{X}(\Delta t), \tag{20}$$

where $\mathbf{U}$ is the instantaneous flow velocity, and the random noise term $\mathbf{X}(\Delta t)$ is drawn from a zero-mean Gaussian with variance $\langle \mathbf{X}(\Delta t)\mathbf{X}(\Delta t) \rangle = 2D\Delta t\mathbf{I}$, where $D$ is the Brownian diffusivity of the particle. In our simulations, we take the diffusion constant to be the mean intranode diffusivity measured in *Holcman et al., 2018*, $D \approx 0.6$ µm² s⁻¹. We include interactions between particles and walls by assuming that particles perfectly reflect off walls (i.e. elastic collisions). The particles are modelled as rigid spheres of diameter 5 nm, and the size of the particle matters only during elastic collisions with the walls of the tubules. As relevant in the limit of low volume fraction, we neglect hydrodynamic interactions between particles and perform ensemble averaging of the trajectories of many independent particles.

When a particle enters a node, we model its dynamics as follows. We consider a particle to have entered a node only if it has reached the end of a tubule, say of length $l$, at which instance we assign the particle to the node point (i.e. the single-point associated with the node in the graph description of the ER network). Although nodes contain a three-dimensional volume, their typical nodal length scale is of the order $R \ll l$, and thus approximating them by point nodes is appropriate on the scale of the whole network. To decide towards which of the connected tubules the particle leaves the node, we estimate the values of the Péclet number Pe in each of the tubules. We define a local Péclet $\mathrm{Pe}_i = U_i R/D$ where $U_i$ is the mean flow velocity through tubule $i$, with $U_i > 0$ for flow out of the node and $U_i \leq 0$ otherwise. We then assume that the particle enters a neighbouring tubule $i$ connected to the node with a probability proportional to $\max(\mathrm{Pe}_i + 1, 0)$. This ensures that we have the expected behaviour in both limits of Pe: at high (positive) Péclet numbers, the probability is proportional to the flow speeds in each of the connected tubules, while at low Péclet the exit of the node is limited by diffusion and thus the exit is equally likely in each tubule.

## Data processing: instantaneous speeds and average edge traversal speeds

During each simulation, we compute the edge traversal speeds as follows. A particle is defined to traverse an edge $(i, j)$ if it travels from node (i.e. junction) $i$ to node $j$, or from $j$ to $i$, without visiting $i$ or $j$ in between. The corresponding edge traversal time is then the time between the arrival at the target node and the most recent departure from the node of origin. The edge traversal speed is naturally defined as the length of the tubule $(i, j)$ divided by the edge traversal time.

The average edge traversal speed associated with an edge $(i, j)$ is then defined as the mean over all edge traversal events across $(i, j)$ of the edge traversal speeds.

In addition, we also compute for each particle the 'instantaneous' speeds defined by $V_n = |\mathbf{X}(t_{n+1}) - \mathbf{X}(t_n)|/\Delta t$, where $t_n = n\Delta t$ with $\Delta t = 18$ ms, which is the same temporal resolution as in the particle tracking carried out in *Holcman et al., 2018*.

## Incorporating slip boundary conditions

The methodology we have detailed thus far assumes no-slip boundary conditions at the tubule walls for the fluid flow. However, the membrane-bound lipids themselves could also flow in response to the

nanoscale luminal flows. This may be modelled by introducing a finite slip boundary condition on the tubule wall. The slip boundary conditions with a slip length $\lambda \geq 0$ at the wall $r = a(z, t)$ are given by

$$u = -\lambda \frac{\partial u}{\partial r},$$ (21)

and $(\mathbf{u} - \frac{\partial a}{\partial t}\mathbf{e}_r) \cdot \mathbf{n} = 0$, with $\mathbf{n} = \frac{\partial a}{\partial z}\mathbf{e}_z - \mathbf{e}_r$, which simplifies to

$$v - \frac{\partial a}{\partial z}u = \frac{\partial a}{\partial t}.$$ (22)

We may then derive the long-wavelength solution for the flow field inside an axisymmetric deforming tubule as follows. Using an ansatz for the axial component $u$ that is motivated by the uniform-radius Poiseuille flow with slip,

$$u(r, z, t) = \frac{2}{1 + \dfrac{4\lambda}{a(z, t)}} \frac{Q(z, t)}{\pi a(z, t)^2} \left[ 1 - \left( \frac{r}{a(z, t)} \right)^2 + \frac{2\lambda}{a(z, t)} \right],$$ (23)

we may solve the incompressibility condition $\frac{1}{r} \frac{\partial}{\partial r}(rv) + \frac{\partial u}{\partial z} = 0$. Regularity at $r = 0$ constrains the integration constant to be zero, yielding

$$
\begin{aligned}
v(r, z, t) = \quad & \frac{1}{1 + \dfrac{4\lambda}{a(z, t)}} \frac{\partial a(z, t)}{\partial t} \frac{r}{a(z, t)} \left[ 2 - \left( \frac{r}{a(z, t)} \right)^2 + \frac{4\lambda}{a(z, t)} \right] \\
& + \frac{2}{1 + \dfrac{4\lambda}{a(z, t)}} \frac{\partial a}{\partial z} \frac{Q(z, t)}{\pi a(z, t)^2} \frac{r}{a(z, t)} \left\{ 1 + 3\lambda - \left( \frac{r}{a(z, t)} \right)^2 \right. \\
& \left. - \frac{\lambda}{a(z, t)} \frac{1}{1 + \dfrac{4\lambda}{a(z, t)}} \left[ \frac{4\lambda}{a(z, t)} + 2 - \left( \frac{r}{a(z, t)} \right)^2 \right] \right\},
\end{aligned}
$$ (24)

which automatically satisfies the boundary condition in **Equation 22**. Note that the mass conservation equations are not affected by the introduction of a slip length.

Using the new solution for $u$, the modified Hagen–Poiseuille expression with slip may be derived as before to be

$$
\begin{aligned}
\Delta p = \quad & -\frac{8\mu}{\pi} \left\{ \frac{L_1}{R^4} + \frac{L}{64\lambda^3(R - b)} \left[ \log\left( \frac{1 + 4\lambda/b}{1 + 4\lambda/R} \right) - \frac{4\lambda}{b}\left( 1 - \frac{2\lambda}{b} \right) + \frac{4\lambda}{R}\left( 1 - \frac{2\lambda}{R} \right) \right] \right\} Q_1 \\
& -\frac{8\mu}{\pi} \left\{ \frac{L_2}{R^4} + \frac{L}{64\lambda^3(R - b)} \left[ \log\left( \frac{1 + 4\lambda/b}{1 + 4\lambda/R} \right) - \frac{4\lambda}{b}\left( 1 - \frac{2\lambda}{b} \right) + \frac{4\lambda}{R}\left( 1 - \frac{2\lambda}{R} \right) \right] \right\} Q_4,
\end{aligned}
$$ (25)

which does recover the no-slip result as $\lambda \to 0$.

These results may then be used to simulate particle transport with slip boundary conditions. In **Figure 3a**, we plot the distributions of average edge traveral speeds obtained from simulations of a C1 network pinching with the original pinch parameters from **Holcman et al., 2018** for four different values of the boundary slip length. In **Figure 3b**, we further display the profiles of the longitudinal flow (**Equation 23**) for different slip lengths with the volume flux fixed.

## Estimate of forces required for pinches

In this section, we derive an order-of-magnitude estimate for the forces required to pinch an ER tubule.

We first estimate the difference in the membrane's elastic energy, $\Delta E = E_{\text{pinched}} - E_{\text{unpinched}}$, between the pinched ($E_{\text{pinched}}$) and unpinched configurations ($E_{\text{unpinched}}$). In the absence of spontaneous curvature, the Helfrich free energy $h$ per unit area of a membrane is given by

$$h = \frac{k_c}{2}(2H)^2 + \bar{k}K,$$ (26)

where $k_c$ and $\bar{k}$ are bending rigidities, $H$ is the mean curvature, and $K$ is the Gaussian curvature (*Helfrich, 1973*). The mean and Gaussian curvatures may be expressed in terms of the principal curvatures $\kappa_1, \kappa_2$ as $H = (\kappa_1 + \kappa_2)/2$ and $K = \kappa_1 \kappa_2$. We take $\kappa_1$ and $\kappa_2$ to be the principal curvatures in the directions normal and parallel, respectively, to the tubule's longitudinal axis.

The dominant contribution to $\Delta E$ is from $E_{\text{pinched}}$, specifically from the region near the centre of the pinch, taken to be at $z = 0$, where the tubule radius is smallest. We have $\kappa_1 = 1/a(z, t)$ and $\kappa_2 = \mathcal{O}(R/L^2)$, yielding $H = 1/2a + \mathcal{O}(R/L^2)$ and $K = \mathcal{O}(R/aL^2)$. The dominant contribution to the membrane energy $E = 2\pi \int_{-L}^{L} ha\, dz$ is therefore from $H$, in a neighbourhood of $z = 0$. Since real pinches do not have kinks, a linear term, for the purposes of estimating membrane energy, is unphysical, and we expect $a(z, t_0) = b_0 + \mathcal{O}(z^2)$ near $z = 0$, where $b_0$ is the pinch radius in the centre of the pinch in the maximally contracted state and $t_0$ denotes a time at which the tubule is in a pinched state. We may therefore write $a(z) = b_0 + Rz^2/L^2$ to obtain an order-of-magnitude estimate of the membrane energy in the pinched state.

Evaluating the integral for $E_{\text{pinched}}$ then yields the leading-order estimate

$$\Delta E \sim k_c \frac{L}{\sqrt{b_0 R}}, \tag{27}$$

We may take $k_c, \bar{k} \sim 50\, k_B T$ ($k_B$ is the Boltzmann constant times and $T$ the room temperature) (*Faizi et al., 2019*), with estimate values $R = 30\,\text{nm}$ and $L = 70\,\text{nm}$. The minimum pinch radius $b_0$ may be estimated to be 10 nm allowing for membrane thickness and incomplete squeeze, yielding

$$\Delta E \sim 8 \times 10^{-19}\,\text{J}. \tag{28}$$

The hydrodynamic contribution to the energy expenditure during a pinch may be calculated as the sum of the dissipation inside the pinch and in the rest of the network (see Energetic cost estimate for contracting peripheral sheets for an analogous calculation for a contracting peripheral sheet). We find that the hydrodynamic contribution is negligible compared to the elastic component of the work done to pinch a tubule.

An estimate for the force required to pinch the tubule may then finally be obtained as $F \sim \Delta E/R \sim 30\,\text{pN}$.

## Derivation of theoretical bounds for active flows driven by pinching tubules

### Advection due to a single pinch

In this section, we calculate an upper bound for the axial distance $\Delta z$ a particle can be advected by the flow produced by an individual pinch. Recall the formula for the volume 'source' due to a pinch,

$$q = -\frac{2}{3}\pi \dot{b}L(R + 2b). \tag{29}$$

Among all possible ways for a particle to be transported by a pinch, an upper bound on the transport distance $\Delta z \le \Delta z_{\text{max}}$ can be reached if all of the following conditions are satisfied: (1) All of the source flows to one side of the pinch (i.e. there is no leakage on the other side); (2) The particle travels outside the pinching region (axial flows within the pinching region produce smaller advective displacements than those outside, as may be verified numerically); (3) The particle travels along the centreline of the tubule (i.e. at twice the cross-section averaged flow velocity, a standard Poiseuille result); (4) The minimum pinch radius $b_0$ (recall from *Figure 10*) is 0.

Under conditions (1) and (2), the cross-section averaged speed corresponding to maximal transport generated from a single pinch can then be computed using *Equation 29* as

$$\pi R^2 \bar{U}(t) = -\frac{2}{3}\pi \dot{b}L(R + 2b). \tag{30}$$

Using condition (3), the position of the particle along the centreline $z(t)$ satisfies then the ordinary differential equation

$$\dot{z} = 2\bar{U}(t) = -\frac{4L}{3R^2}\dot{b}(R + 2b). \tag{31}$$

Integrating this equation from $t = 0$ (start of contraction, with $b(0) = R$) to $T$ (end of contraction, with $b(T) = b_0$) then leads to

$$\Delta z = \frac{8L}{3} \left( 1 - \frac{b_0}{2R} - \frac{b_0^2}{2R^2} \right). \tag{32}$$

The value of $\Delta z$ is maximised when $b_0 = 0$ (i.e. when condition (4) holds), yielding the upper bound

$$\Delta z \leq \Delta z_{\max} = \frac{8}{3}L. \tag{33}$$

## Extension to nonlinear interactions between two pinches

An isolated pinch is only capable of reciprocal motions. The simplest system capable of producing non-reciprocal motions is illustrated in *Figure 5* and consists of two pinch sites arranged in series near the midpoint of a long horizontal tubule.

We may calculate an upper bound on the net particle displacement that can be achieved after both pinch sites pinch exactly once. We make the assumptions (2)–(4), and in addition, allow pinches to spend extended amounts of time in their completely closed state; any deviation from these assumptions will result in net transport that is further reduced.

How can we maximise the positive (i.e. rightward in *Figure 5*) particle displacements induced by the contractions, and minimise the magnitude of the negative (leftward) displacement induced by relaxations? As in the figure, let us denote the pinch on the left 'pinch 1' and that on the right 'pinch 2'. Pinch 1 produces maximum positive displacement (magnitude $4L/3$, i.e. half of the optimal value from *Equation 33*) when pinch 2 is completely open, that is, when the hydrodynamic resistance to its right is minimal. Pinch 2 then produces maximum positive displacement (of magnitude $8L/3$, i.e. the optimal value in *Equation 33*) when pinch 1 is completely closed and all pinch-induced flow continues to be directed rightwards. Similarly, pinch 1 produces minimal negative displacement when pinch 2 is completely closed (zero average displacement, since no flow is then allowed to escape to the right of pinch 2), while pinch 2 then produces a negative displacement of magnitude $4L/3$ to the left when pinch 1 is completely open. These optima can be achieved by the non-reciprocal sequence of motions illustrated in *Figure 5*: close pinch 1, close pinch 2, open pinch 1, and open pinch 2. The coordination between two pinches can therefore be used to generate the net displacement of $8L/3$ equal to the theoretical upper bound from *Equation 33*.

## Modelling of alternative flow generation mechanisms

We have described in detail our modelling of a network driven by the pinching of tubules. In this study, we explore two other flow generation mechanisms: the contraction of tubular junctions and the contraction of ER sheets. Our model for pinching tubules is readily generalised to account for these mechanisms, as we describe below.

### Experimental estimates of junction volumes

From fluorescence microscopy images of ER networks, we measure the fluorescence intensity of junctions (i.e. the number of pixels in a junction multiplied by the mean intensity per pixel). We can then translate this intensity into an estimate of junction volume, assuming that intensity is directly proportional to volume. In order to calibrate the fluorescence intensity, we use our measurements for tubules. Specifically, we use the measured intensities of tubules and their known volumes, obtained by measurements of tubule lengths and the assumption that they are cylinders of radius 30 nm, in order to determine the proportionality constant to convert between pixel intensity and volume. Our new measurement of the intensities of the junctions then allows us to obtain estimates of their volumes; we obtain twelve values, ranging between 0.0020 and 0.0081 $\mu m^3$, with a mean of 0.045 $\mu m^3$ and an SD of 0.0021 $\mu m^3$.

### Mathematical modelling of contractions of tubular junctions

To include the contribution of tubular junctions into the model (*Figure 1c*), we assume that in addition to the pinch sites along tubules, each tubular junction pinches independently of other tubules

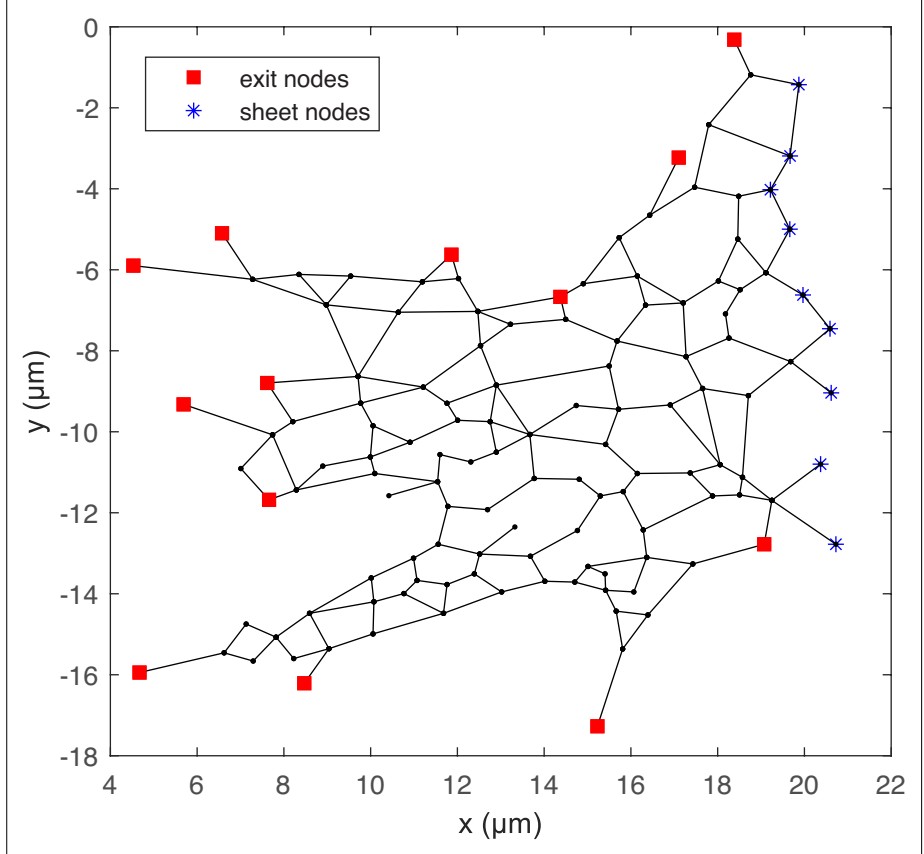

**Figure 12.** Illustration of the C1 network from *Figure 9* with $M_1 = 13$ exit nodes (red squares) and $M_2 = 9$ perinuclear sheet nodes (blue asterisks).

and other tubular junctions. Given a junction, we assume that it expels the same volume $\Delta V$ of fluid during a contraction for all its pinches (and takes in the same volume when it relaxes). Each pinch is assumed to create a sinusoidal flow source, so a pinch lasting for a time $2T$ produces a flow rate $S(t) = \Delta V \sin(\pi t/T)\pi/2T$ at a time $t$ measured from the beginning of the pinch, where the numerical factors in $S(t)$ are chosen such that the volume expelled during a contraction is indeed $\int_0^T S(t)\,dt = \Delta V$. We accommodate the flows from the junctions mathematically by modifying our K1 equations, *Equation 17*, to allow the normal nodes to also carry non-zero sources (as opposed to just the pinch nodes, as was the case before); the other equations in the model remain unchanged.

## Mathematical modelling of contractions of perinuclear sheets

In the tubule-pinching model, we included $M$ exit nodes located towards the exterior of the network and through which flow could enter and exit the system in order to conserve mass. To account for the connection to a perinuclear sheet (*Figure 1f*), we now assign a number of these exit nodes, denoted $M_2 < M$, to be 'sheet nodes', i.e. nodes which are directly connected to a perinuclear sheet, so that a number $M_1 = M - M_2 > 0$ of exit nodes remain. This is illustrated in *Figure 12*, where we show the C1 network from *Figure 9* with both sheet nodes (blue asterisks) and exit nodes (red squares).

A sheet contraction + relaxation lasting a time $2T$ produces a total source $S_{\text{sheet}}(t) = V_{\text{sheet}}\pi \sin(\pi t/T)/2T$ at a time $t$ from the beginning of the pinch, where again, the integral of $S_{\text{sheet}}$ over a contraction gives a volume $V_{\text{sheet}}$ of fluid expelled by the sheet.

Similarly to the mathematical model with tubular pinches only, our independent variables are the $E$ tubule fluxes, the $M_1$ sources at the exit nodes, and the $M_2$ sources at the sheet nodes. As before, the K1 (*Equation 17*) and K2 (*Equation 18*) equations give us $E + 1$ equations. The requirement (analogous to *Equation 19*) that the exit nodes are at the same mechanical pressure gives us $M_1 - 1$ equations. We make the additional assumption that the sheet nodes are all at the same mechanical

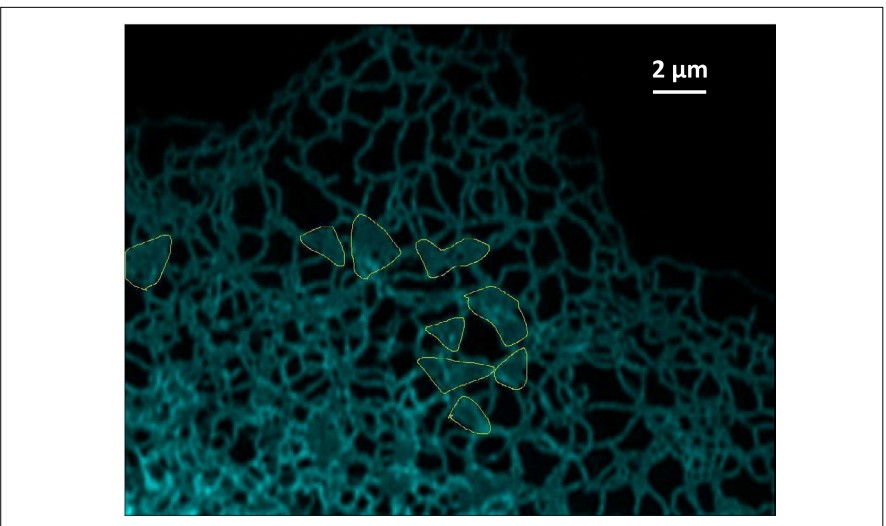

**Figure 13.** Estimation of areas of peripheral sheets, taken as the regions encircled in yellow.

pressure (i.e. that they are connected to the same reservoir), which provides an additional $M_2 - 1$ equations. Requiring that the sources at the sheet nodes sum to the prescribed flow rate $S_{\text{sheet}}$ yields one additional equation, so we again have a system of $E + M$ independent linear equations.

## Experimental estimates of volumes of peripheral sheets

To model the contraction of peripheral sheets (*Figure 1d*), we need to estimate their contained volumes. Using the open-source image analysis software Fiji (*Schindelin et al., 2012*), we identify nine regions roughly occupied by peripheral sheets in a microscopy image of an ER network; this is illustrated in yellow in *Figure 13*. We then measure their areas (in µm²) and convert them to volumes by multiplication with the diameter of a tubule, taken as 60 nm, assuming that the effective thickness of a sheet is equal to the tubular diameter. From our nine data values, we finally obtain a mean sheet volume of 0.12 µm³ and an SD of 0.04 µm³.

## Energetic cost estimate for contracting peripheral sheets

The dominant energy expenditure in deforming a pinching tubule is in creating the large membrane curvatures at the narrowest sections of the pinch, and much less work is done against the small volumes of fluid displaced (Estimate of forces required for pinches). A contracting peripheral sheet, however, displaces a relatively large volume of fluid without attaining the extreme curvatures required in a pinching tubule. We therefore expect, intuitively, that the hydrodynamic contribution will dominate

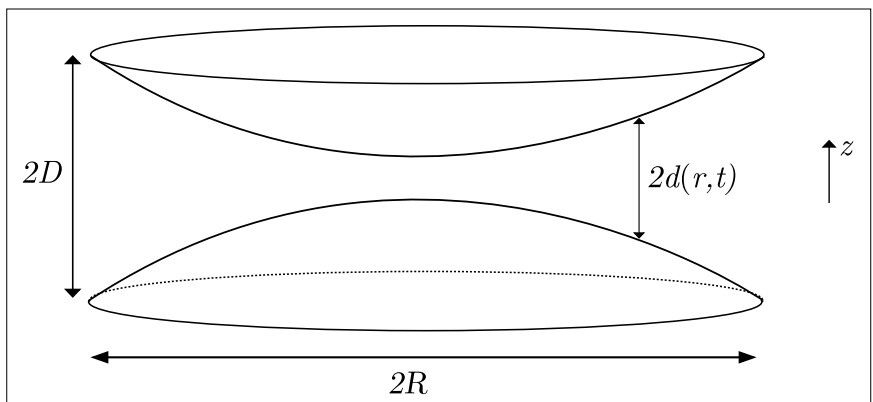

**Figure 14.** Mathematical idealisation of two contracting peripheral sheets as two paraboloids for the purpose of computing an order-of-magnitude estimate of the energy expended to contract a peripheral sheet.

the energy expenditure budget. We now explicitly show this by estimating both elastic and hydrodynamic contributions.

To derive an order-of-magnitude estimate of the energetic cost, we consider an idealisation of a peripheral sheet consisting, in the relaxed state, of two parallel circular membranes $r < R$ located at $z = \pm D$, and in the fully contracted state, of two paraboloids at $z = \pm Dr^2/R^2$ (see *Figure 14*). The membranes deform as paraboloids between these two states, and we denote the 'vertical' distance between the two membranes as $d(r, t)$. We take $R = 0.8$ µm, the value consistent with a sheet thickness $2D = 60$ nm and the mean sheet volume (in the fully relaxed state) of $2\pi R^2 D = 0.12$ µm³ (see estimation in Experimental estimates of volumes of peripheral sheets).

To estimate the elastic contribution, we present an argument similar to the one carried out in Estimate of forces required for pinches. In the fully contracted state, the principal curvatures of one membrane scales as $\sim 2D/R^2$, so the Gaussian curvature $K \sim 4D^2/R^4$ and the mean curvature $H \sim 2D/R^2$, yielding the Helfrich energy density $h \sim k[2H^2 + K] \sim 12kD^2/R^4$, where $k$ is the typical bending rigidity. The total bending energy of the fully contracted membrane is then $E \sim \pi R^2 h \sim 2 \times 10^{-20}$ J. In the relaxed state, the membranes are flat and have zero bending energy. Thus, the total energy required to contract the two membranes can be estimated as

$$\Delta E \approx 4 \times 10^{-20} \text{ J}. \tag{34}$$

To estimate the hydrodynamic contribution, we first note that the work done instantaneously by the contracting sheet against the fluid, of dynamic viscosity $\mu$, say, is the sum of dissipation rate in the sheet itself and the dissipation rate in the rest of the ER network outside the sheet due to the sheet-induced flows. Any work done against the fluid outside the ER network is neglected since flows decay over short length scales across the network.

We first consider the contribution from within the sheet. We may use lubrication theory to estimate the flows inside a contracting peripheral sheet since the sheet is fairly flat. The leading-order lubrication flow between the contracting membranes (located at $z = \pm d(r, t)$) is in the radial direction, and given by $u_r = -\frac{1}{\mu}\frac{\partial p}{\partial r}(d^2 - z^2)$, where the unknown radial pressure gradient $\frac{\partial p}{\partial r}$ may be calculated from mass conservation $2\pi r \int_{-d}^{d} u_r \, dz = -\frac{\partial}{\partial t}\left(\int_0^r 4\pi r' d(r', t) \, dr'\right)$, thus yielding a radial velocity

$$u_r = -\frac{3\int_0^r r' \dot{d} \, dr'}{2rd^3}(d^2 - z^2). \tag{35}$$

The dominant contribution to the rate-of-strain tensor is $e_{rz} \approx \frac{1}{2}\frac{\partial u_r}{\partial z} \sim \frac{3R}{2DT}$, where $T \sim 0.5$ s is the contraction duration (in the main text we considered a value of $T$ of 2.5 or 5 times larger than the experimentally measured pinch duration; here we take 5). Therefore, the total work done over a contraction scales as the total dissipation rate times the contraction duration $T$, and we scale

$$W_{\text{inside}} \sim 2\mu \frac{9R^2}{4D^2T^2} \times 2\pi R^2 DT = \frac{9\pi\mu R^4}{DT}. \tag{36}$$

Taking the viscosity of the intra-luminal fluid to be 10 times that of water, this is computed to be

$$W_{\text{inside}} \sim 8 \times 10^{-18} \text{ J}. \tag{37}$$

The dissipation rate in the network outside the sheet may be estimated by calculating the total dissipation rate in an idealised network consisting of 'generations' of tubules, each of length 1 µm, with the first generation consisting of three tubules connected to the peripheral sheet, and each tubule in the $i^{\text{th}}$ generation branching out into three tubules of the $(i + 1)^{\text{th}}$ generation ad infinitum. The dissipation rate due to a Poiseuille flow of flux $q$ inside a tubule of length $l$ and radius $R$ may be calculated to be

$$\mathcal{D}_{\text{tube}} = \frac{8\mu q^2 l}{\pi R^4}. \tag{38}$$

Denoting by $Q$ the total volume flux created by the contracting sheet, we may calculate the dissipation rate in the network of tubules by summing the above expression across all the tubules as follows. The $i^{\text{th}}$ generation of tubules is comprised of $3^i$ tubules each carrying a flux of $Q/3^i$. Summing the total dissipation rates across all generations gives

$$\mathcal{D}_{\text{network}} = \sum_{i=1}^{\infty} 3^i \frac{8\mu l}{\pi R^4} \left(\frac{Q}{3^i}\right)^2 = \frac{4\mu Q^2 l}{\pi R^4}. \tag{39}$$

We may scale $Q$ as $Q \sim V_s/2T$, recalling our assumption that each peripheral sheet contraction expels half of the volume $V_s$ contained in the sheet (which is consistent with the paraboloidal membrane profile we have taken for the fully contracted sheet). The total dissipation over the contraction duration $T$ in this network of tubules is therefore

$$W_{\text{outside}} \sim \frac{\mu V_s^2 l}{\pi R^4 T}. \tag{40}$$

Scaling the sheet volume with its mean value $V_s \sim 0.12 \ \mu m^3$, we may compute this to be

$$W_{\text{outside}} \sim 10^{-16} \ \text{J}. \tag{41}$$

The total work done during a contraction is therefore $W = W_{\text{inside}} + W_{\text{outside}} \sim 10^{-16}$ J. Given that the energy released by the hydrolysis of one ATP molecule is of the order of $10^{-19}$ J (*Bray, 2001*), we thus estimate that each contraction of a peripheral sheet would require on the order of 1000 molecules of ATP.

## Acknowledgements

EL and PHH are supported by the European Research Council under the European Union's Horizon 2020 research and innovation program (Grant No. 682754 to EL) and by the Cambridge Trust. EA is supported by the UK Dementia Research Institute [award number UK DRI-2004] which receives its funding from UK DRI Ltd, funded by the UK Medical Research Council, Alzheimer's Society and Alzheimer's Research UK.

## Additional information

### Funding

| Funder | Grant reference number | Author |
|---|---|---|
| Horizon 2020 Framework Programme | 682754 | Eric Lauga |
| UK Dementia Research Institute | DRI-2004 | Edward Avezov |
| Cambridge Trust | 303328966 | Pyae Hein Htet |

The funders had no role in study design, data collection, and interpretation, or the decision to submit the work for publication.

### Author contributions

Pyae Hein Htet, Data curation, Software, Formal analysis, Validation, Investigation, Visualization, Methodology, Writing - original draft, Writing – review and editing; Edward Avezov, Resources, Supervision, Funding acquisition, Investigation, Project administration, Writing – review and editing; Eric Lauga, Conceptualization, Resources, Supervision, Funding acquisition, Validation, Investigation, Methodology, Project administration, Writing – review and editing

### Author ORCIDs

Pyae Hein Htet ⓘ https://orcid.org/0000-0001-5068-9828
Edward Avezov ⓘ https://orcid.org/0000-0002-2894-0585
Eric Lauga ⓘ https://orcid.org/0000-0002-8916-2545

Reviewer #1 (Public review): https://doi.org/10.7554/eLife.93518.3.sa1
Author response https://doi.org/10.7554/eLife.93518.3.sa2

## Additional files

### Supplementary files
- MDAR checklist

### Data availability
Modelling code is available in the Zenodo repository.

The following dataset was generated:

| Author(s) | Year | Dataset title | Dataset URL | Database and Identifier |
|---|---|---|---|---|
| Htet PH | 2023 | stevenhtet/ERFLOW: Fluid mechanics of luminal transport in actively contracting endoplasmic reticulum: modelling code (v1.0) | https://doi.org/10.5281/zenodo.7521244 | Zenodo, 10.5281/zenodo.7521244 |

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
