## [Editor Report · eLife Assessment]

This work explores the physical principles underlying fluid flow and luminal transport within the endoplasmic reticulum. Its **important** contribution is to highlight the strong physical constraints imposed by viscous dissipation in nanoscopic tubular networks. In particular, the work presents **convincing** evidence based on theoretical analysis that commonly discussed mechanisms such as tubular contraction are unlikely to be at the origin of the observed transport velocities. As such, it will be of relevance to cell biologists and physicists interested in organelle dynamics. As this study is solely theoretical and deals with order of magnitude estimates, its main conclusions await experimental validation.

---

## [Referee Report · Reviewer #1 (Public review)]

Theoretical principles of viscous fluid mechanics are used here to assess likely mechanisms of transport in the ER. A set of candidate mechanisms are evaluated, making good use of imaging to represent ER network geometries. Evidence is provided that contraction of peripheral sheets provides a much more credible mechanism than contraction of individual tubules, junctions or perinuclear sheets.

The work has been conducted carefully and comprehensively, making good use of underlying physical principles. There is good discussion of the role of slip; sensible approximations (low volume fraction, small particle size, slender geometries, pragmatic treatment of boundary conditions) allow tractable and transparent calculations; clear physical arguments, including an analysis of energy budgets, provide useful bounds; stochastic and deterministic features of the problem are well integrated.

---

## [Author Response]

The following is the authors’ response to the original reviews.

**Reviewer #1 (Public Review):**
Theoretical principles of viscous fluid mechanics are used here to assess likely mechanisms of transport in the ER. A set of candidate mechanisms is evaluated, making good use of imaging to represent ER network geometries. Evidence is provided that the contraction of peripheral sheets provides a much more credible mechanism than the contraction of individual tubules, junctions, or perinuclear sheets.The work has been conducted carefully and comprehensively, making good use of underlying physical principles. There is a good discussion of the role of slip; sensible approximations (low volume fraction, small particle size, slender geometries, pragmatic treatment of boundary conditions) allow tractable and transparent calculations; clear physical arguments provide useful bounds; stochastic and deterministic features of the problem are well integrated.

We thank the reviewer for their positive assessment of our work.

There are just a couple of areas where more discussion might be warranted, in my view.(1) The energetic cost of tubule contraction is estimated, but I did not see an equivalent estimate for the contraction of peripheral sheets. It might be helpful to estimate the energetic cost of viscous dissipation in generated flows at higher frequencies.

This is a good point. We have now included an energetic cost estimate for the contractions of peripheral sheets in the revised manuscript.

The mechanism of peripheral sheet contraction is unclear: do ATP-driven mechanisms somehow interact with thermal fluctuations of membranes?

The new energetic estimates in the revision might help constrain possible hypotheses for the mechanism(s) driving peripheral sheet contraction, and suggest if a dedicated ATP-driven mechanism is required.

(2) Mutations are mentioned in the abstract but not (as far as I could see) later in the manuscript. It would be helpful if any consequences for pathologies could be developed in the text.

We are grateful for this suggestion. The need to rationalise pathology associated with the subtle effects of mutations of ER-morphogens is indeed pointed out as one factor motivating the study of the interplay between ER structure and performance. In the revised manuscript, we have included a brief discussion potentially linking the malfunction of ER morphogens to luminal transport, referencing freshly published findings.

**Reviewer #2 (Public Review):**
Summary:This study explores theoretically the consequences of structural fluctuations of the endoplasmic reticulum (ER) morphology called contractions on molecular transport. Most of the manuscript consists of the construction of an interesting theoretical flow field (physical model) under various hypothetical assumptions. The computational modeling is followed by some simulations.Strengths:The authors are focusing their attention on testing the hypothesis that a local flow in the tubule could be driven by tubular pinching. We recall that trafficking in the ER is considered to be mostly driven by diffusion at least at a spatial scale that is large enough to account for averaging of any random flow occurring from multiple directions [note that this is not the case for plants].

We thank the reviewer. Indeed, the trafficking in the ER was historically presumed to be driven by passive diffusion but this has been challenged by recent findings suggesting that the transport may also involve an active super-diffusional component (the short-lasting flows). These findings include: the dependence of ER luminal transport on ATP-derived energy observed in the historical and recent publications cited here; fast and directional single-particle motion; and a linear scaling of photoactivated signal arrival times with distance. On a larger scale, indeed, the motion can be seen as a faster effective diffusion, as there is no persistent circulatory directionality of the currents.

Weaknesses:The manuscript extensively details the construction of the theoretical model, occupying a significant portion of the manuscript. While this section contains interesting computations, its relevance and utility could be better emphasized, perhaps warranting a reorganization of the manuscript to foreground this critical aspect.Overall, the manuscript appears highly technical with limited conclusive insights, particularly lacking predictions confirmed by experimental validation. There is an absence of substantial conclusions regarding molecular trafficking within the ER.

We sought to balance the theoretical/computational details of our model with the biophysical conclusions drawn from its predictions. Given the model's complexity and novelty, it was essential to elucidate the theoretical underpinnings comprehensively, in order to allow others to implement it in the future with additional, or different, parameters. To maintain clarity and focus in the main text, we have judiciously relegated extensive technical details to the methods section or supplementary materials, and divided the text into stand-alone section headings allowing the reader to skip through to conclusions.

The primary focus of our manuscript is to introduce and explore, via our theoretical model, the interplay between ER structure dynamics and molecular transport. Our approach, while *in silico*, generates concrete predictions about the physical processes underpinning luminal motion within the ER. For instance, our findings challenge the previously postulated role of small tubular contractions in driving luminal flow, instead highlighting the potential significance of local flat ER areas—empirically documented entities—for facilitating such motion.

Furthermore, by deducing what type of transport may or may not occur within the range of possible ER structural fluctuations, our model offers detailed predictions designed to bridge the gap between theoretical insight and experimental verification. These predictions detail the spatial and temporal parameters essential for effective transport, delineating plausible values for these parameters. We hope that the model’s predictions will invite experimentalists to devise innovative methodologies to test them. We have introduced text edits to the revised version to clarify the reviewer’s point as per the detailed comments below.

**Recommendations for the authors:**

**Editor comments (Recommendations For The Authors):**
The two reviewers have different opinions about the strengths and weaknesses of this work. The editors do believe that this work is a valuable contribution to the field of ER dynamics and transport, and could stimulate experiments.

We thank both reviewers and the editors for the time and care they have invested in reviewing our manuscript.

Nevertheless, discussing further the role of diffusion vs. advection in ER luminal transport, including conflicting values of measured diffusion coefficients, would be valuable. For instance, it is possible that the active contraction-driven mechanism results in an effective diffusion over a long time, which could be quantified and compared to experiments.

In our study we focus on tubule-scale transport because the statistics of transport at this scale have been measured and the origins of the observed transport is an outstanding problem. We already know from Holcman et al. (2018) that transport at the tubule scale involves an active, possibly advective, component beyond passive molecular diffusion. Although we do touch briefly on a network-scale phenomenon in our section on mixing/content homogenisation, our main focus is on trying to understand tubule-scale transport. We agree that a substantial exploration of effective diffusion over a network scale would be of value and increase the breadth of our paper, we feel that this is beyond the scope of the current paper. We believe the “conflicting” diffusion coefficients, in fact, characterise motion at different time and length scales: the global diffusion coefficient pointed out to us in the reviews may pertain to network-scale *effective* diffusion over long time scales, but this is different to the Brownian motion on the scale of tubules/tubular junctions relevant to our *in silico* model.

**Reviewer #1 (Recommendations For The Authors):**
I congratulate the authors on their work and do not have any substantial further recommendations, beyond two minor points.(1) Before (13), say "Using the expression (7) for Q_2, ..."(2) Typo on p.25: "principal" rather than "principle" (two instances)

We thank the reviewer for spotting these and have addressed both points.

**Reviewer #2 (Recommendations For The Authors):**
Here are some specific comments:(1) Insufficient Influence of ER Tubule Contraction:The conclusion regarding weak fluid flows generated by ER tubule contractions may seem obvious. It would be more intriguing if the authors explored conditions necessary to achieve faster flows, such as those around 20 µm/s, within tubules.

We agree these are important conditions to explore and it is extensively covered in Fig. 4e-f, which show that tubule contraction sites the length of entire tubules and occurring at 5 and 10 times the experimentally measured rates produce mean average edge traversal speeds exploring otherµconceivable scenarios. of ~30 and 60 m/s respectively. These pinch parameters seemed unlikely and motivated

(2) Limited Impact of ER Network Geometry:The comparison across different ER network structures seems insufficiently documented. A comparison between distal and proximal ER from the nucleus could provide deeper insights.

We have added text in the new paragraph 4 of the introduction to better articulate the core principles of the ER’s structural elements. As established by historical EM and light microscopy, the ER is universally composed of tubules, with 3-way junctions, and small (peripheral) or large perinuclear sheets. We establish that the specific shaping of these elements influences the nanofluidics we investigate here. While the proportion of these elements may vary across different cell types and cellular regions, the fundamental structure, and therefore the impact on local mobility remains consistent. Our categorisation of the ER into its elements reflects these ubiquitous components, allowing us to analyse the impact of shaping at the relevant scale, covering the perinuclear and peripheral ER.

(3) Ineffectiveness of Tubule Junction Contraction:The study's negative result on ER tubule junction contraction's impact on molecular exchange may not capture broad interest without experimental validation. Conducting experiments to test this hypothesis could strengthen the study.

We agree that experimental testing of this prediction in the future, when appropriate tools become available to correlate molecular motion speed and fast contractions of nanoscopic tubular junctions, will be needed for its validation.

(4) Potential Role of Peripheral Sheets:While the speculation on the contraction of peripheral ER sheets is intriguing, further experimental investigation is warranted to validate this hypothesis, especially considering the observed slow diffusion in ER sheets.

We agree with the reviewer that our study is theoretical in nature and on the necessity of further experimental investigation before we are able to make a definitive conclusion on peripheral sheets.

In summary, while the study underscores the complexity of ER morphology dynamics and its implications for molecular transport, its novelty and broad implications seem limited. Given its reliance on computational simulations and dense theoretical language, submission to a computational journal could be more appropriate. In addition, given there is an absence of substantial conclusions regarding molecular trafficking within the ER, publication in a specialized journal of fluid mechanics or physics may be appropriate.Comments:- The manuscript is hard to read. There is no smooth transition from Figure 1 to Figure 2.

To smoothen the transition, we edited the text at the beginning of results and added a reference there to the introductory Fig. 1.

- Figure 8 serves no purpose. To make the text easier, C0, C1, C2... should be presented in Figure 2 and merged with Figure 10 with a table summarizing the information of these networks. It is not clear why 5 networks are needed. They look similar. Could you add the number of nodes per network?

We have now merged Fig 8 and Fig 10 from the previous version into one figure (which is now Fig 9). We have also added information about the number of nodes and added a sentence in the manuscript to clarify that it showcases the source data used to model/reconstruct realistic ER structures.

- Figure 13: seems out of contex. What is the message? The ER does not show any large flow--from early FRAP and recent photoactivation - the material seems to diffuse at long distances made by few tubules.

Fig 13 (now Fig 12 in the revised version) does not illustrate any flow. Its purpose is to illustrate the computational methodology used to simulate flows and transport due to contraction of perinuclear sheets. (Note that we have spotted and fixed a small but important typo in the caption: “peripheral” →”perinuclear”.) It is worth noting that FRAP provides a relative estimate of mobility but contains no information as to the mode of motion. Whether the motion is diffusive or otherwise must be presumed in FRAP analyses and this presumption then can be used to extract metrics such as the diffusion coefficient. Photoactivation analyses suffer from the same limitation but analysis of how photoactivated signal arrival times scale with distance was recently suggested as a workaround. These measurements suggest a superdiffusive ER transport (https://doi.org/10.1016/j.celrep.2024.114357). Although a different approach used in a recent preprint to photoactivation signal analysis suggests that at long-distances transport can be approximated as diffusion (https://doi.org/10.1101/2023.04.23.537908), improved measurement in the future would be needed to address the seeming discrepancies.

- Figure 1: what is the difference between a and b? How do you do your cross-section? This probability needs a drawing at least to understand how you define it.

We expanded the explanation in the third last paragraph of Section I.

- Figure 2: this manuscript is not a review. It is not clear why part of a figure is copied and pasted from another manuscript. It should be removed. Are the authors using the quantification [peaks of different color]? Where? The title should be given to explain each panel.

We have chosen to keep the inset, which was not in the main text of the cited paper but its supplementary information, and provides a direct benchmark for our work.

Why the mean flow in a is stochastic? With large excursion for large values? Could you plot the Fourrier or spectrogram so we can understand the frequencies? Are there regular patterns of bursts?

The mean (i.e. cross-sectionally averaged) flow is stochastic because the pinching events are random (more precisely, they follow a Poisson distribution, as explained in the paper). Large excursions are rare and caused by interactions between pinches. We have prescribed the distributions of pinch durations and frequencies as per experimentally measured distributions and we do not expect to recover from a Fourier analysis more information than we have prescribed.

What do we learn from the fit of Fig 2b-c? Is it a constant flow?

The conclusion of Fig 2b-c is that the in silico simulation model based on the pinching tubule hypothesis produces solute transport, as quantified by the instantaneous particle speeds (Fig 2b) and the average edge traversal speeds (Fig 2c), that is much weaker than experimentally measured. This is one of the main results of our paper and explained in Section IIA, paragraph 3. Fig 2a tells us that the flow is not constant (flows in this system can only be generated transiently, with directionality persistence considered unlikely).

Figure 14: Estimating of the area is unclear. The legend is largely insufficient.Why did the authors report only nine regions of contractions? Is it so rare? How many samples have they used? Nine among how many?

Thank you; the details of area estimation are included in the main text, in Section I.4. The nine regions are an arbitrary selection of a sample we deemed representative of this phenomenon.

- Abstract: this is misleading, it should start by explaining that diffusion is the consensus of trafficking in the ER.- "the content motion in actively contracting nanoscopic tubular networks" is misleading. We should recall that this is an assumption that has not been proven.

The current abstract is a succinct summary of the question in scope and results. The sentence highlighted by the referee specifically refers to the model we study in the paper; we modified it in order to remove any ambiguity and to make clear that we are testing a proposed mechanism. We also point out that although the biological origins of the tubule contractions or their effects on solute transport have not been established, these contractions have been documented.

Minor comments:Introduction: "Thus past measurements indicate that the transport of proteins in ER is not consistent with Brownian motion" is misleading. You should explain that this depends on the time scale. At large timescale, diffusion is a coarse-grained description and is actually accurate from FRAP and photoactivation data [see J. Lippincott-Schwartz publications over the past 20 years].The super-diffusion [9] "A photoactivation chase technique also measured a superdiffusive behaviour of luminal material spread through the ER network [9]." This is not clear and is probably due to an artifact of measurements or interpretation.

We thank the reviewer for this comment. We expanded in paragraph 2 of the introduction to better reflect the state of knowledge around this point.

Page 2 "Strocytes" does not exist you may be meant "Astrocytes".

Thank you; typo fixed.

Page 5: The value of the flow seems incompatible with previous literature ~ 20 mu m/s. Again where 0.6 is found? Where in [7]: if there is no diffusion in the tubule, why compare with 0.6 mu m^2/s? The global diffusion coefficient is much higher ~ 5 mu m^2/s.

Supplementary Figure 3b of µHolcman *et al.,* Nat Cell Biol, 2018 (Ref. [7] in the unrevised version: The value of 0.6 m^2/s is the intranodal diffusion coefficient reported empirically in our article), for ER in COS-7 cells. Motion inside the tubule would in general consist of a combination of advection and diffusion; since the same fluid occupies the tubules and the m^2/s as the diffusion coefficient in tubules as well. The experiments in Holcman *et al.* (2018) µ junctions, and the junction sizes are similar to the tubule diameters, it is reasonable to take 0.6 does not mention diffusion inside tubules because (i) the study reports a dominantly advective (or at least active) transport across tubules (the driving mechanism of which remains unknown) but this does not mean diffusion is not there as well; and (ii) the time resolution in these experiments are too low to capture the fine details of solute motion inside tubules, and the transport is captured only as “jumps” between junctions. We point out also that the higher global diffusion coefficient may pertain to network-scale effective diffusion over long time scales, which is different to the Brownian motion at the scale of tubules/tubular junctions relevant to our _in silic_o model.

Page 5: "The distributions of the average edge traversal speeds appeared insensitive to ER structure variations for both pinching-induced and exclusively diffusion transport." is rather trivial. Similar to "the presumed pinching parameters would be inadequate to facilitate ER luminal material exchange."

These sentences, and the surrounding text, report the observed outcome of our numerical simulations: pinch-induced transport statistics has little variation across different ER geometries, and pinching does not facilitate luminal content mixing. This conclusion was not clear to us without running the simulation, and hence we deemed it nontrivial and relevant to comment on.

Page 7: The authors mention that they could measure "typical edge traversal speed of 45 µm/s".

I am not aware of such a measurement. Could they explain where this number comes from?

These measurements were reported in Supplementary Figure 3b (bottom right) of Holcman *et al.* (2018) and are for the ER in a COS-7 cell. The main figure Fig. 2g reports analogous measurements for COS-7 cells because the tubule contraction data reported in this work measurements (mean speed 20 m/s) for a HEK-293 cell. We have worked with the speed pertains to COS-7 cells.

A contraction that leads to 3.9 mu m/s over a distance of a few microns would be interesting. Is this a prediction of the present model?

Yes, as stated in Section II, C paragraph 2. The present model with the experimentally measured averages for the tubule contraction parameters does indeed predict that a particle, in the absence of diffusion, is transported by a single tubule contraction at a *maximum* speed of 3.9 µm/s over 0.19 µm.